# ADAPTIVE THRESHOLD SAMPLING FOR FAST NOISY SUBMODULAR MAXIMIZATION

## ABSTRACT

We address the problem of submodular maximization where objective function $f : 2^U \to \mathbb{R}_{\geq 0}$ can only be accessed through i.i.d noisy queries. This problem arises in many applications including influence maximization, diverse recommendation systems, and large-scale facility location optimization. We propose an efficient adaptive sampling strategy, called `Confident Sample` (CS), that is inspired by algorithms for best-arm-identification in multi-armed bandit, which significantly improves sample efficiency. We integrate CS into existing approximation algorithms for submodular maximization, resulting in algorithms with approximation guarantees arbitrarily close to the standard value oracle setting that are highly sample-efficient. We propose and analyze sample-efficient algorithms for monotone submodular maximization with cardinality and matroid constraints, as well as unconstrained non-monotone submodular maximization. Our theoretical analysis is complemented by empirical evaluation on real instances, demonstrating the superior sample efficiency of our proposed algorithm relative to alternative approaches.

## 1 INTRODUCTION

Submodularity is a property of set functions that arises in many applications such as cut functions in graphs Balkanski et al. (2018), coverage functions Bateni et al. (2017), data summarization objectives Tschiatschek et al. (2014), information theoretic quantities such as mutual information Iyer et al. (2021), and viral marketing in social networks Kempe et al. (2003). A function $f : 2^U \to \mathbb{R}_{\geq 0}$ defined over subsets of the universe $U$ of size $n$ is submodular if for all $X \subseteq Y \subseteq U$ and $u \notin Y$, $f(Y \cup \{u\}) - f(Y) \leq f(X \cup \{u\}) - f(X)$. In addition, in many applications of submodular functions $f$ is monotone (Tschiatschek et al., 2014; Iyer et al., 2021; Kempe et al., 2003), meaning that for all $X \subseteq Y \subseteq U$, $f(X) \leq f(Y)$. Proposed algorithms for submodular optimization typically are assumed to have value oracle access to $f$. That is, $f$ is a black box that can be queried for any $X \subseteq U$, and the value of $f(X)$ is returned Nemhauser et al. (1978); Badanidiyuru & Vondrák (2014); Balkanski et al. (2019a); Buchbinder et al. (2015).

However, in many optimization scenarios, we can only make noisy queries from some random distribution to estimate the objective. For example, in applications such as diversified recommender system Yue & Guestrin (2011); Hiranandani et al. (2020), data summarization with human feedback Singla et al. (2016), influence maximization Kempe et al. (2003); Wen et al. (2017), feature selection tasks Krause & Guestrin (2005), querying the exact value of $f$ is unrealistic, and instead a more realistic assumption is that we can query $f$ subject to some random noise. In particular, we assume that the noisy sampling of $f$ is random and is i.i.d sub-Gaussian, which is also referred to as bandit feedback in the submodular bandit literature Singla et al. (2016); Chen et al. (2017). In a related setting, submodular optimization algorithms that leverage the multilinear extension $F$ of the function $f$ may only be able to access $F$ via i.i.d noisy random samples and this is a major bottleneck in terms of the efficiency of these algorithms Calinescu et al. (2011); Badanidiyuru & Vondrák (2014). In this setting, the common approach is to use existing submodular optimization algorithms and apply the fact that the objective can be evaluated to arbitrary precision by taking sufficiently many samples and applying concentration inequalities in order to achieve a fixed-precision (see Section 2) approximation of the objective function Kempe et al. (2003); Calinescu et al. (2011). However, modern massive datasets demand algorithms that are as efficient as possible in terms of runtime, and

in the case of submodular optimization algorithms, the main computation time bottleneck for the above approach would be the noisy queries to $f$.

Motivated by the above, our main insight is that an algorithm doesn't necessarily need to approximate $f$ with such fine precision at every query in order to find a solution with an approximation guarantee comparable to the exact value oracle setting. Instead, we propose methods of adaptively approximating the function f based on decisions that the algorithm must make, with an emphasis on minimizing the total number of noisy queries. Methods of efficient sampling in order to determine the best action is related to the best-arm identification in submodular bandit, where the objective is that to identify a super-arm (subset of the universe) with comparable approximation ratio in as few samples as possible (Audibert et al., 2010; Singla et al., 2016). Therefore our algorithmic contributions and analysis are inspired by ideas used in best-arm-identification in the bandit setting. In particular, the contributions of the paper are as follows:

(i) We propose the adaptive sampling algorithm `Confident Sample` (CS) in Section 3, which can be used to determine if the mean of a random variable $X$ is approximately above or below a given threshold $w$ with high probability, in relatively few random samples. Intuitively, the required number of samples is inversely proportional to the gap between $EX$ and $w$, and therefore we can significantly decrease the number of samples relative to the fixed-precision approach by sampling less when the gap is large. CS is related but significantly different from algorithms used for best-arm-identification in bandit, as we explain in detail in Section 4. CS is used as a subroutine for all proposed algorithms for submodular maximization problems in the paper, and as a result the proposed algorithms exhibit an improved sample complexity compared with fixed-precision approximation.

(ii) We address the problem of Monotone Submodular Maximization with Cardinality constraint (MSMC) in Section 4, which is defined to find the set $\arg\max\{f(X) : X \subseteq U, |X| \leq \kappa\}$. We prove two results for the proposed `Confident Threshold Greedy` algorithm (CTG), Theorem 3 and Theorem 4. Theorem 3 is demonstrated to achieve an improved sample complexity compared with that of the related work of Singla et al. (2016), while achieving the same approximation guarantee. Theorem 4 is proved to achieve a better sample complexity compared with the sampling before-hand approach in the application of influence maximization.

(iii) In Section 5, the algorithm `Confident Continuous Threshold Greedy` (CCTG) is proposed and analyzed for the problem of Monotone Submodular Maximization with Matroid constraint (MSMM). MSMM is to find the solution of $\arg\max_{S \subseteq \mathcal{M}} f(S)$, where $\mathcal{M}$ is a matroid defined on subsets of the ground set $U$. CCTG accesses the multilinear extension of $f$ via noisy samples, since the multilinear extension can be difficult to compute in general Calinescu et al. (2011); Badanidiyuru & Vondrák (2014). In particular, we demonstrate that CCTG has an improved sample complexity compared with the one proposed in Badanidiyuru & Vondrák (2014).

(iv) In Section E, we propose `Confident Double Greedy` (CDG) for Unconstrained Submodular Maximization (USM). The goal is to find a subset $S \subseteq U$ that maximizes $f(S)$ where $f$ is not necessarily monotone. The theoretical guarantee on sample complexity is presented in Theorem 15 in the appendix.

(v) Finally, as a demonstration of our approach, we experimentally analyze CTG on instances of noisy data summarization and influence maximization. We compare CTG to several alternative methods including the algorithm of Singla et al. (2016) which is discussed in more detail in Section 1.1 and in the appendix. CTG is demonstrated to be a practical choice that can save many samples relative to alternative approaches.

## 1.1 RELATED WORK

Approximation algorithms for the maximization of a submodular objective function subject to various constraints have been extensively studied in the literature Nemhauser et al. (1978); Badanidiyuru & Vondrák (2014); Mirzasoleiman et al. (2015); Calinescu et al. (2011) with the assumption of oracle access to $f$. The runtime of these algorithms is generally measured in queries to $f$ as this is the main bottleneck (see Section A for a more comprehensive discussion on the runtime of algorithms for various submodular optimization problems).

While there are many works assuming value oracle access to $f$, algorithms developed assuming noisy access to $f$ are relatively less explored Horel & Singer (2016); Singla et al. (2016); Hassidim & Singer (2017); Qian et al. (2017); Crawford et al. (2019); Huang et al. (2022). One related setting to ours is that we have noisy access to $f$, but this noise is *persistent* Horel & Singer (2016); Hassidim & Singer (2017); Qian et al. (2017); Crawford et al. (2019); Huang et al. (2022). Our noisy setting departs from this direction in that the noisy feedback is random and repeated samples should be taken to diminish the noise. Another related but different setting is that of stochastic submodular optimization Karimi et al. (2017); Staib et al. (2019); Özcan & Ioannidis (2023) which assumes the optimization objective $f$ is the expectation over some unknown distribution over a set of monotone submodular functions. Therefore a sample average function can be built, which is also monotone and submodular, and algorithms run on it. In contrast, in our setting, it is only assumed that we can sample noisy queries at each subset $X \subseteq U$. The algorithm ExpGreedy of Singla et al. (2016) is for a noisy setting identical to ours and is developed for the MSMC problem specifically. ExpGreedy also incorporates an adaptive sampling approach. In particular, their algorithm combines the standard greedy algorithm with the best arm identification problem found in combinatorial bandit literature Chen et al. (2014). Their approach is still very different from ours, and an extensive comparison of our algorithms and results with Singla et al. (2016) are presented in the appendix, as well as an experimental comparison in Section 6.

The intuition behind CS is similar to the best-arm-identification problem in the multi-armed bandit literature Kalyanakrishnan et al. (2012). Both the algorithm LUCB of Kalyanakrishnan et al. (2012) and CS share a common underlying intuition: they leverage the difference between expectations to reduce the number of noisy queries required. In LUCB, this difference is between the expectation of the optimal arm and other arms, while in CS, it is between the expectation of the input variable and the threshold value $w$.

## 2 PRELIMINARY DEFINITIONS AND NOTATIONS

In this section, we lay the groundwork definitions and notations for the remainder of the paper. Throughout this paper, we assume $f : 2^U \to \mathbb{R}_{\geq 0}$ is submodular. $U$ is the ground set of size $n$. Let us denote the marginal gain of adding element $u \in U$ to a set $X \subseteq U$ as $\Delta f(X, u)$, i.e., $\Delta f(X, u) := f(X \cup \{u\}) - f(X)$.

We first define the noisy model of access to $f$. In particular, given any subset $X \subseteq U$ and $u \in U$, independent samples can be taken from the distribution $\mathcal{D}(X, u)$ to obtain noisy evaluations of $\Delta f(X, u)$. In this paper, we denote the random variable following the distribution of $\mathcal{D}(X, u)$ as $\widetilde{\Delta f}(X, u)$. We assume the following properties about the distribution $\mathcal{D}(X, u)$: (i) $\mathbb{E}[\widetilde{\Delta f}(X, u)] = \Delta f(X, u)$; and (ii) $\widetilde{\Delta f}(X, u)$ are bounded in the range of $[0, R]$ for all $X, u$ (or in some results, they are assumed to be $R$-sub-Gaussian).[1] In addition, in applications where instead we have noisy queries directly to $f$ instead of the marginal gain, this also satisfies our setting (see Section A in the appendix of the supplementary material for more details).

Below we describe three different types motivating examples of our noisy setting and illustrate the value of $R$ on these instances.

1. **Diversified recommender systems with human feedback.** In this problem, the goal is to select a subset of items to recommend to users. The objective function is the total number of expected clicks by the users, typically defined by the cascading linear submodular bandit model Hiranandani et al. (2020). In this setting, the objective function is computed in expectation and can only be estimated through noisy feedback from the users. A noisy sample corresponds to querying a person for feedback, and samples are i.i.d. On a related note, the crowdsourced image source summarization considered by Singla et al. (2016) follows a similar setting, where noisy samples correspond to human feedback. The maximum value of feedback is then bounded by 1. Therefore can be set to be $1/2$ for Theorem 1 and 1 for Theorem 2.

---

[1] A random variable that is bounded within the interval $[0, R]$ can be demonstrated to be $R/2$ sub-Gaussian. Consequently, the assumption of a random variable being sub-Gaussian is more general than that of boundedness.

2. **Multi-linear extension.** This setting specifically applies to our Algorithm CCTG, which is our continuous algorithm that uses the multilinear extension of $f$ to achieve an improved approximation guarantee for the matroid constraint. The multilinear extension is commonly used in submodular optimization algorithms, and is defined as $\mathbf{F}(\mathbf{x}) = \sum_{S \subseteq U} \prod_{i \in S} x_i \prod_{j \notin S} (1 - x_j) f(S)$ where $\mathbf{x} \in [0, 1]^n$. Notice that obtaining the true value of the multi-linear extension requires an exponential number of queries, therefore the proposed algorithms often require sampling to approximate function values. Noisy queries for the true value of the multilinear extension can be obtained by taking i.i.d. samples of sets, further described in Section 5 of our paper. On this instance, the noisy marginal gain is bounded by the maximum singleton value, so we can set $R$ to be $\max_{s \in U} f(s)$.

3. **Stochastic submodular maximization.** Our problem setup covers the class of stochastic submodular maximization (SSM) problems. The objective function of an SSM problem can be expressed as $f(S) = \mathbf{E}_\gamma[f_\gamma(S)]$. To solve this problem, we would need to approximate the function value $f$ by taking samples of $f_\gamma(S)$ from the distribution of $\gamma$. Since the distribution of $\gamma$ doesn't change, the sampling of the function $f$ for each fixed $S$ is i.i.d. A specific application of this problem is the **influence maximization** problem, where the objective function is the expected number of nodes influenced in the graph by a seed set $S$. This problem has wide applications in social network analysis. (For a detailed definition of influence maximization, please refer to the Appendix D.1). Another set of problems that can also be solved by SSM is the large-scale weighted sum submodular maximization problem where the objective can be expressed as $f(S) = \sum_{i=1}^N w_i f_i(S)$. Here $N$ is very large and $\sum_{i=1}^N w_i = 1$. Examples of this problem include large-scale facility location optimization. In this problem, the cost of accurately evaluating a problem would be high, but we can estimate $f(S)$ by sampling the index $I \in [N]$ with probability $w_i$ and then $f(S) = \mathbf{E}_I[f_I(S)]$.

Next, we present the definition of fixed $\epsilon$-approximation and multi-linear extension.

**Fixed $\epsilon$-approximation.** Given any random variable $X$, an estimate $\widehat{X}$ is a *fixed $\epsilon$-approximation* of $X$ if $\mathbf{E}X - \epsilon \leq \widehat{X} \leq \mathbf{E}X + \epsilon$. Notice that for any $X$ that is $R$-sub-Gaussian, we can take $O\left(\frac{R^2}{\epsilon^2} \log \frac{1}{\delta}\right)$ samples and the sample average is a fixed $\epsilon$-approximation of $X$ with probability at least $1-\delta$ by an application of Hoeffding's Inequality (Lemma 19 in the appendix in the supplementary material).

**Multi-linear extension**. For any submodular objective $f$, the multi-linear extension of $f$ is defined as $\mathbf{F}$, i.e., $\mathbf{F}(\mathbf{x}) = \sum_{S \subseteq U} \prod_{i \in S} x_i \prod_{j \notin S} (1 - x_j) f(S)$ where $\mathbf{x} \in [0, 1]^n$. Here we define $S(\mathbf{x})$ to be a random set that contains each element $i \in U$ with probability $x_i$, then by definition, we have that $\mathbf{F}(\mathbf{x}) = \mathbf{E}[f(S(\mathbf{x}))]$.

## 3 CONFIDENT SAMPLING ALGORITHM

In this section, we propose and analyze the `Confident Sample` (CS) algorithm. CS is used in order to determine if the expected value of a random variable $X$ is approximately above or below a threshold value with high probability. CS works for any random variable that is $R$-sub-Gaussian (see Theorem 1) or bounded in the range of $[0, R]$ (see Theorem 2). In Sections 4, E, and 5, we show that CS is useful as a subroutine for a variety of submodular maximization algorithms where we only have noisy access to the marginal gains.

We now describe CS. CS takes as input failure probability $\delta \in \mathbb{R}_{>0}$, threshold error parameter $\epsilon \in \mathbb{R}_{>0}$, a threshold value $w \in \mathbb{R}_{>0}$, the unknown distribution $\mathcal{D}_X$ of the random variable $X$, and the sub-Gaussian parameter $R$. CS iteratively takes at most $N_1$ samples from $\mathcal{D}_X$, while maintaining a sample average and a confidence interval. In particular $\widehat{X}_t$ is the sample average after taking $t$-th samples of $X$, i.e., $\widehat{X}_t = \frac{1}{t} \sum_{i=1}^t X_i$ where $X_i$ is the $i$-th random sample of $X$. The confidence region, after taking the $t$-th sample of $X$, is a shrinking region $[\hat{X}_t - C_t, \hat{X}_t + C_t]$ around $\widehat{X}_t$ that reflects where CS is almost certain that the true value of $\mathbf{E}X$ has to be. We leave the exact definition of both $C_t$ and $N_1$ until Theorems 1 and 2 for reasons that will become clear. Once the lower bound of the confidence region crosses $w - \epsilon$, or the upper bound crosses $w + \epsilon$, CS completes and returns true

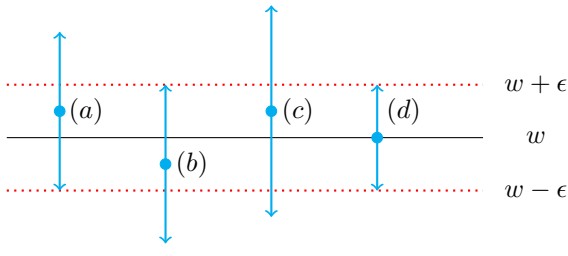

Figure 1: An illustration of the various states of CS. The blue dots depict the values of $\widehat{X}_t$, while the surrounding blue lines depict the confidence region $[\widehat{X}_t - C_t, \widehat{X}_t + C_t]$. Once the region looks like (a), CS will return true. In (b), CS will return false. In (c), CS will continue sampling to reduce the width of the confidence region. Finally, in (d) CS has taken $N_1$ samples resulting in an $\epsilon$-additive approximation.

---

**Algorithm 1:** Confident Sample (CS)

---

1: **Input:** $w, \epsilon, \delta, \mathcal{D}_X, R$
2: **for** $t = 1, 2, ...N_1$ **do**
3: $\widehat{X}_t \leftarrow$ updated sample mean after taking $t$-th sample from $\mathcal{D}_X$
4: $C_t \leftarrow$ updated confidence interval
5: **if** $\widehat{X}_t - C_t \geq w - \epsilon$ **then**
6:  **return true**
7: **else if** $\widehat{X}_t + C_t \leq w + \epsilon$ **then**
8:  **return false**
9: **end if**
10: **end for**
11: **if** $\widehat{X}_t \geq w$ **then**
12: **return true**
13: **else**
14: **return false**
15: **end if**

---

or false respectively. Note that the CS algorithm differs significantly from the fixed-$\epsilon$ approximation approach commonly used in the submodular optimization literature, such as Algorithm 2 in Fahrbach et al. (2019). A detailed discussion of this distinction is provided in Section C.1 of the appendix.

We now state our first main result for CS in Theorem 1 below. The second item of Theorem 1 states that with high probability, CS will correctly return the answer to whether $\boldsymbol{E}X$ is approximately above or below the input threshold $w$. The first item states that, in the worst case, CS takes $O(R^2 \log(1/\delta)/\epsilon^2)$ samples from $\mathcal{D}_X$ to return true or false no matter what the value of $\boldsymbol{E}X$ is. However, the further the value of $\boldsymbol{E}X$ is from $w$, as reflected by $\phi$, the fewer samples CS needs to make a decision. Figure 2 illustrates how the sample complexity changes with the increase of gap function $\phi$ in the result of Theorem 1.

The details of the proof of Theorem 1 can be found in Section C.2 of the supplementary material.

**Theorem 1.** *For any random variable $X$ that is $R$-sub-Gaussian, if we define $N_1 = 2R^2/\epsilon^2 \log \frac{4}{\delta}$, and $C_t = R\sqrt{\frac{2}{t} \log \frac{8t^2}{\delta}}$, then the algorithm* Confident Sample *achieves that with probability at least $1 - \delta$*

 *1. CS on input $(w, \epsilon, \delta, \mathcal{D}_X, R)$ takes at most the minimum between*

$$\left\{ \frac{2R^2}{\epsilon^2} \log\left(\frac{4}{\delta}\right), \frac{8R^2}{\phi_X^2} \log\left(\frac{16R^2}{\phi_X^2}\sqrt{\frac{2}{\delta}}\right) \right\}$$

 *noisy samples, where $\phi_X = \frac{\epsilon + |w - \mathbb{E}X|}{2}$.*

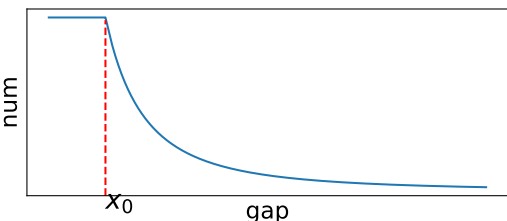

Figure 2: A plot to illustrate how the number of samples taken by CS (num) changes with the gap function $\phi_X$ (see Theorem 1). There exists some $x_0$ such that when $0 < \phi_X \leq x_0$, the required number of samples is $\frac{R^2}{2\epsilon^2} \log \frac{2}{\delta}$ (the left side in the sample complexity result in Theorem 1). When $\phi_X > x_0$, the right-hand side in Theorem 1 is the minimum and the sample complexity of the algorithm decreases fast as $\phi_X$ increases.

    *2. If CS returns true, then $\boldsymbol{E}X \geq w - \epsilon$. If CS returns false, then $\boldsymbol{E}X \leq w + \epsilon$.*

Here we provide explanation for the result of sample complexity in the second point of Theorem 1. The term on the left-hand side, $\frac{2R^2}{\epsilon^2} \log\left(\frac{4}{\delta}\right)$, represents the number of samples required to approximate $X$ within $\epsilon$-distance with probability, i.e., $|X - \boldsymbol{E}X| \leq \epsilon$. This corresponds to case (d) in Figure 1, and is the number of samples that the fixed $\epsilon$-approximation would take. Such a large number of samples is only necessary when $\boldsymbol{E}X$ is close to the threshold, and therefore many samples are needed to see if it is above or below the threshold. Importantly, this value can be obtained without adaptive sampling.

The value on the right-hand side comes from the adaptive sampling, and it is the number of samples required to shrink the confidence interval just enough so that we can conclude whether $\mathbb{E}X$ is approximately above or below the threshold, and it depends on how far $\mathbb{E}X$ is from the threshold i.e. the value of $\phi$ (since a larger gap allows for a wider confidence interval upon stopping and thus fewer samples. ). This latter value cannot be computed before we start sampling, and is a result of the adaptive sampling where we do not know how many samples we will take initially. This corresponds to cases (a) and (b) in Figure 1.

Our second result, Theorem 2, is related to Theorem 1 but instead of an additive approximation error (i.e. $\boldsymbol{E}X \geq w - \epsilon$ or $\boldsymbol{E}X \leq w + \epsilon$), the error is a combination of multiplicative and additive. The intuition behind using this result is that in many submodular algorithms that require the thresholding procedure, the threshold decreases exponentially which allows the multiplicative error. On the other hand, in the case where $R$ can be as large as $n$, the result in Theorem 2 can be more sample efficient. In order to get Theorem 2, a different definition of the confidence radius $C_t$ as well as the maximum number of samples $N_1$ is needed. Theorem 2 is proven in the supplementary material in Section C.3.

**Theorem 2.** *For any random variable $X$ that is bounded in the range of $[0, R]$, if we define $C_t = \frac{3R}{t\alpha} \log(\frac{8R^2}{\delta})$, and $N_1 = \frac{3R}{\alpha\epsilon} \log(\frac{4}{\delta})$ where $\alpha$ is an additional parameter that controls the multiplicative error rate, the algorithm* Confident Sample *achieves that with probability at least $1 - \delta$*

    *1. CS on input $(w, \epsilon, \delta, \mathcal{D}_X, R)$ takes at most the minimum between*

$$\left\{ \frac{3R}{\epsilon\alpha} \log\left(\frac{4}{\delta}\right), \frac{12R}{\alpha\phi'_X} \log\left(\frac{12R}{\alpha\phi'_X}\sqrt{\frac{8}{\delta}}\right) \right\}$$

    *noisy samples, $\phi'_X = \frac{\epsilon - \alpha\boldsymbol{E}X + |w - \mathbb{E}X|}{2}$.*
    *2. If the output is true, then $(1 + \alpha)\boldsymbol{E}X \geq w - \epsilon$. If the output is false, then $(1 - \alpha)\boldsymbol{E}X \leq w + \epsilon$.*

## 4  MONOTONE SUBMODULAR MAXIMIZATION

In this section, we address the MSMC problem under the noisy setting, where we assume the noisy sampling of the marginal gain $\Delta f(S, s)$ is $R$-sub-Gaussian for any $S \subseteq U$ and $s \in U$. Necessary definitions and notations are first given in Section 2. We propose two algorithms Confident

---

**Algorithm 2:** `Confident Threshold Greedy (CTG)`

1: **Input:** $\epsilon, \delta, \alpha$
2: $N_2 \leftarrow 2R^2 \log(6n/\delta)/(\epsilon^2)$
3: **for all** $s \in U$ **do**
4:    $\hat{f}(s) \leftarrow$ sample mean over $N_2$ samples from $\mathcal{D}(\emptyset, s)$
5: **end for**
6: $d := \max_{s \in U} \hat{f}(s),$
7: $w \leftarrow d, S \leftarrow \emptyset$
8: **while** $w > \alpha d/\kappa$ **do**
9:    **for all** $u \in U$ **do**
10:      **if** $|S| < \kappa$ **then**
11:        thre = `Confident Sample` $(w, \epsilon, \frac{2\delta}{3nh(\alpha)}, \mathcal{D}(S, u), R)$
12:        **if** thre **then**
13:          $S \leftarrow S \cup \{u\}$
14:        **end if**
15:      **end if**
16:    **end for**
17:    $w = w(1 - \alpha)$
18: **end while**
19: **return** $S$

---

`Threshold Greedy (CTG)` and `Confident Threshold Greedy2 (CTG2)` for this problem. A detailed description of `CTG` is given in Section 4.1. The approximation and sample complexity guarantees of `CTG` and `CTG2` are presented in Theorem 3 and Theorem 4 in Section 4.2. For `CTG2`, the algorithm description and pseudocode are provided in Section D.3 of the appendix.

## 4.1 ALGORITHM DESCRIPTION OF `CTG`

Here we describe `Confident Threshold Greedy (CTG)`. `CTG` is based on the algorithm `Threshold Greedy (TG)` of Badanidiyuru & Vondrák (2014) which is for MSMC with an exact value oracle. Pseudocode for `CTG` can be found in Algorithm 2.

The algorithm `CTG` takes as input a parameter $\alpha \in (0, 1)$. `CTG` proceeds in $O(\log(\kappa/\alpha)/\alpha)$ *rounds*, where each round corresponds to a value of $w$. The threshold $w$ is first set to $d$, which is an $\epsilon$-additive approximation of the maximum singleton value with high probability. In particular, $d$ satisfies that with probability at least $1 - \delta/3$, $\max_{s \in U} f(s) + \epsilon \geq d \geq \max_{s \in U} f(s) - \epsilon$. During each round, `CTG` iterates through all elements in $U$. Since for each $S$ and $u$, the noisy query to the marginal gain $\Delta f(S, u)$ is $R$-sub-Gaussian, `CTG` can use `CS` as the subroutine to determine whether to include $u$ to the solution set $S$. Here $h(\alpha) = \frac{\log(\kappa/\alpha)}{\alpha}$. The worst-case query complexity $N_1$ and confidence interval $C_t$ in `CS` are defined as in Theorem 1.

## 4.2 THEORETICAL GUARANTEES AND ANALYSIS

The main result of `CTG` is the Theorem 3 below.

**Theorem 3.** *Suppose the noisy marginal gain of any subset $S \subseteq U$ and element $s \in U$ is $R$-sub-Gaussian, then `CTG` makes at most $n \log(\kappa/\alpha)/\alpha$ calls of `CS`. In addition, with probability at least $1 - \delta$, the following statements hold:*

- *The exact function value of the output solution set $S$ satisfies that $f(S) \geq (1 - e^{-1} - \alpha)f(OPT) - 2\kappa\epsilon$;*
- *Each call of `CS` on input $(w, \epsilon, \frac{2\delta}{3nh(\alpha)}, \mathcal{D}(S, u), R)$ takes at most the minimum between*

$$\left\{ \frac{8R^2}{\phi^2(S, u)} \log\left( \frac{16R^2\sqrt{\frac{3nh(\alpha)}{\delta}}}{\phi^2(S, u)} \right), \frac{2R^2}{\epsilon^2} \log\left( \frac{6nh(\alpha)}{\delta} \right) \right\}$$

*and noisy samples. Here $OPT$ is an optimal solution to the MSMC problem, $\phi(S, u) = \frac{\epsilon + |w - \Delta f(S,u)|}{2}$, and $h(\alpha) = \frac{\log(\kappa/\alpha)}{\alpha}$.*

The proof and analysis of Theorem 3 are deferred to Section D.2 in the appendix. We make a comparison of the theoretical guarantees between our results and those of `ExpGreedy` in Singla et al. (2016), which combines the standard greedy algorithm with the best arm identification algorithm used in bandit literature. The detailed discussion is provided in Section B in the appendix. Here we briefly summarize the results as follows.

First of all, we consider the runtime. Since `ExpGreedy` requires updating the confidence interval for all the elements and two sorting of all elements each time a noisy query is taken, the required runtime is $O(n \log n)$. However, both `CTG` and `EPS-AP` have more efficient runtime complexity and require only one update of the confidence interval in Line 4 and two comparisons in Line 5 and 7 in `CS`, which is only $O(1)$ in computation.

Next, we consider sample complexity. `ExpGreedy` is based on the standard greedy algorithm where each iteration takes at most $O\left(n\kappa' R^2 \min\left\{\frac{4}{\Delta_{\max}^2}, \frac{1}{\epsilon^2}\right\} \log\left(\frac{R^2 \kappa n \min\{\frac{4}{\Delta_{\max}^2}, \frac{1}{\epsilon^2}\}}{\delta}\right)\right)$ samples, which depends on the gap $\Delta_{\max}$ between the top two marginal gains. Therefore, the sample complexity can be sensitive to the small difference between top elements. However, our results depend on $\phi$, which only depends on the difference between and is thus more robust. When $\Delta$ and $\phi$ are in the same order, the average sample complexity per marginal gain in `CTG` is better than `ExpGreedy`. In addition, The total evaluated marginal gain in `CTG` is smaller compared with `ExpGreedy`.

Next, we present the theoretical guarantee of `CTG2` (Algorithm 3, provided in Appendix D.3) in Theorem 4, the proof of which is deferred to Section D.2 in the appendix.

**Theorem 4.** *Suppose the noisy marginal gain of any subset $S \subseteq U$ and element $s \in U$ is bounded in $[0, R]$, `CTG2` makes at most $3n \log(\kappa/\alpha)/\alpha$ calls of `CS`. In addition, with probability at least $1 - \delta$, the following statements hold:*

- *The exact function value of the output solution set $S$ satisfies that $f(S) \geq (1 - e^{-1} - \alpha)f(OPT) - 2\kappa\epsilon$;*
- *Each call of `CS` on input $(w, \epsilon, \frac{2\delta}{3nh'(\alpha)}, \mathcal{D}(S, u), R)$ takes at most the minimum between*

$$\left\{\frac{9R}{\epsilon\alpha} \log\left(\frac{6nh'(\alpha)}{\delta}\right), \frac{36R}{\alpha\phi'(S, u)} \log\left(\frac{36R}{\alpha\phi'(S, u)}\sqrt{\frac{12nh'(\alpha)}{\delta}}\right)\right\}$$

*noisy samples. Here $OPT$ is an optimal solution to the MSMC problem, $\phi'(S, u) = \frac{\epsilon - \alpha\Delta f(S,u)/3 + |w - \Delta f(S,u)|}{2}$, and $h'(\alpha) = \frac{3}{\alpha} \log\left(\frac{3\kappa}{\alpha}\right)$.*

Here the term $nh(\alpha)$ and $nh'(\alpha)$ in Theorem 1 and Theorem 2 represents the total number of calls to the `CS` algorithm respectively. The result of sample complexity is derived by setting the failure probability $\delta$ in Theorem 1 and Theorem 2 to be the reciprocal of the total number of calls to CS, (i.e. the number of marginal gain queries) multiplied by $\delta'$. This adjustment ensures that, via a union bound, the overall algorithm succeeds with a probability of at least $1 - \delta$.

Notice that the sample complexity in Theorem 4 has a dependence of $O(R)$ concerning the order of the parameter $R$, while the sample complexity result in Theorem 3 is $O(R^2)$ in the order of $R$. Consequently, in some applications such as influence maximization, where $R$ can be as large as the size of ground set $n$, Theorem 4 has an advantage in sample complexity compared with Theorem 3. Another related method is the classic sampling-before-hand approach as described in Section D.1 in the appendix. Compared with this approach, `CTG2` has improved sample complexity and is more practical since in real-world scenarios, it might be impossible to store all the graph data and obtain the sampling of an entire graph. (see Section D.1 for more details.)

## 5 CONTINUOUS THRESHOLD GREEDY WITH NOISY QUERIES

In this section, we consider the problem of Monotone Submodular Maximization with a Matroid constraint (MSMM) assuming noisy access to $f$. More specifically, we assume that for any set $S \subseteq U$ and element $s \in U$, the noisy marginal gain $\widetilde{\Delta}f(S, s)$ is bounded in $[0, R]$. In many applications, even with access to an exact oracle for $f$, **F** is not able to be evaluated exactly due to the inherent randomness in $S(\mathbf{x})$ in the definition of **F** (see Section 2), so we can only make noisy queries

to $\mathbf{F}$. In addition, our results hold even for the case that only noisy access to $f$ is provided. We propose the `Confident Continuous Threshold Greedy` (CCTG) algorithm for MSMM, which leverages the continuous multilinear extension $\mathbf{F}$ of the submodular function $f$ to obtain an approximation guarantee arbitrarily close to the best possible result of $1 - 1/e$.

We now describe `CCTG`, the pseudocode of which is deferred to Algorithm 5 in Section F of the appendix. Let $\kappa$ to denote the rank of the matroid, and let $S(\mathbf{x})$ be a random set that contains each element $i \in U$ with probability $x_i$ The `CCTG` algorithm initializes a solution in the origin, $\mathbf{x} = \mathbf{0}$. Then at each step, `CCTG` selects a subset of coordinates $B$ to increment by a predetermined step size $\epsilon$. The set of coordinates $B$ is chosen by the subroutine algorithm `Decreasing-Threshold Procedure` (DTP), which is described in Algorithm 6. Here the parameters $N_1$ and $C_t$ in the subroutine algorithm `CS` are defined as in Theorem 2 with the multiplicative error parameter $\alpha$ set to be $\epsilon/3$. After the `CCTG` is complete, we process the fractional solution $\mathbf{x}$ with the swap rounding procedure in Vondrák et al. (2011) to obtain the final solution set $S$.

**Theorem 5.** *CCTG makes at most $\frac{3n}{\epsilon^2} \log \frac{3\kappa}{\epsilon}$ calls of CS. In addition, with probability at least $1 - \delta$, the following statements hold:*

- *The output fractional solution $\boldsymbol{x}$ achieves the approximation guarantee of $\boldsymbol{F}(\boldsymbol{x}) \geq (1 - e^{-1} - 2\epsilon)f(OPT) - R\epsilon$.*
- *Each call of CS on input $(w, \frac{\epsilon R}{2\kappa}, \frac{\delta\epsilon}{2nh'(\epsilon)}, \mathcal{D}_X, R)$ requires at most the minimum between*

$$\left\{ \frac{18\kappa}{\epsilon^2} \log\left( \frac{8nh'(\epsilon)}{\delta\epsilon} \right), \frac{36R}{\epsilon\phi''_X} \log\left( \frac{144R}{\epsilon\phi''_X} \sqrt{\frac{nh'(\epsilon)}{\delta\epsilon}} \right) \right\}$$

*noisy queries to the marginal gain. Here $OPT$ is an optimal solution to the MSMM problem. $\phi''_X = \frac{\frac{\epsilon R}{2\kappa} - \epsilon\mathbb{E}X/3 + |w - \mathbb{E}X|}{2}$ , and $h'(\epsilon) = \frac{3}{\epsilon} \log\left(\frac{3\kappa}{\epsilon}\right)$.*

The proof of Theorem 5 is deterred to Appendix F. Besides, we discuss and compare our results in Theorem 5 with the Accelerated Continuous Greedy algorithm in Badanidiyuru & Vondrák (2014) in Section F.1. Here we briefly summarize the results as follows: First of all, in the case where we have exact access to the value oracle, the sample complexity of `CCTG` is better than Accelerated Continuous Greedy algorithm in Badanidiyuru & Vondrák (2014) while both algorithms achieve the approximation ratio of $1 - 1/e - O(\epsilon)$. Second, in the case where $\Delta f$ is noisy, as long as the upper bound on the noisy marginal gain $R$ is less than $f(OPT)$, the sample complexity and approximation ratio remains the same. Therefore, the assumption of access to noisy marginal gain does not lead to additional sample complexity or worse approximation ratio when compared to the scenario with an exact value oracle.

## 6 APPLICATIONS AND EXPERIMENTS

In this section, we conduct an experimental evaluation of our algorithm `CTG` on instances of MSMC with noisy marginal gain evaluations. In particular, we consider instances of the noisy data summarization application, which is described in Section H.1.1 in the appendix. Synthetic noise is introduced into marginal gain queries by adding a zero-mean Gaussian random variable with $\sigma = 1.0$ ($\sigma$ is the standard deviation) to the exact value of marginal gain. Therefore, parameter $R = 1.0$. Our experiments are conducted on a subset of the Delicious dataset of URLs that are tagged with topics Soleimani & Miller (2016), and subsets of the Corel5k dataset of tagged images Duygulu et al. (2002). We give more details about the datasets we use in the appendix in the supplementary material. We additionally consider the influence maximization problem in the appendix in the supplementary material. The setup of our experiments is described in Section 6.1, while our results are presented in Section 6.2.

### 6.1 EXPERIMENTAL SETUP

We now describe the setup of our experiments. In addition to our algorithm `CTG`, we compare the following alternative approaches to noisy MSMC: (i) The fixed $\epsilon$ approximation ("EPS-AP") algorithm; (ii) Two special case of the algorithm `ExpGreedy` of Singla et al. (2016) "EXP-GREEDY" and "EXP-GREEDY-K" with the parameter $k'$ in `ExpGreedy` set to be $k' = 1$ and $k' = \kappa$

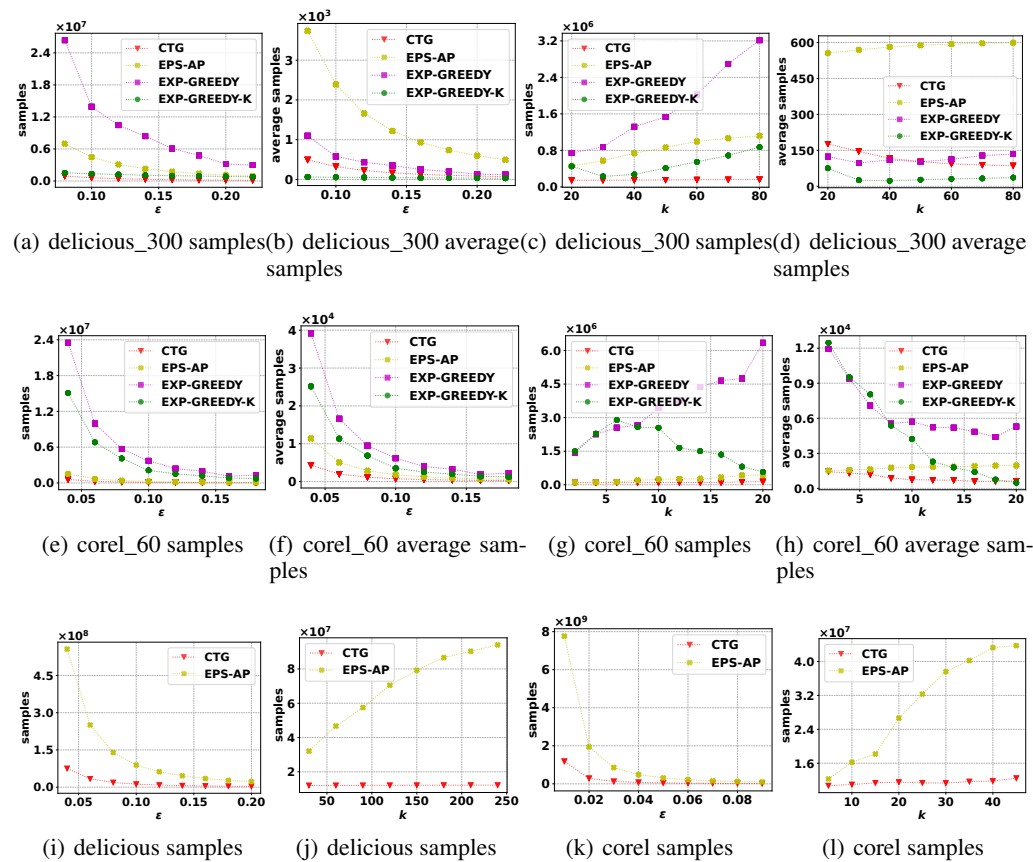

(a) delicious_300 samples (b) delicious_300 average (c) delicious_300 samples (d) delicious_300 average
samples                              samples

(e) corel_60 samples (f) corel_60 average samples (g) corel_60 samples (h) corel_60 average samples

(i) delicious samples (j) delicious samples (k) corel samples (l) corel samples

Figure 3: The experimental results of running different algorithms on instances of data summarization on the delicious URL dataset ("delicious", "delicious_300") and Corel5k dataset ("corel", "corel_60").

respectively. More details about the three algorithms can be found in the appendix. We evaluate `CTG` and `EPS-AP` on all the datasets. However, `EXP-GREEDY` and `EXP-GREEDY-K` have greater runtime as discussed in the appendix in the supplementary material, and so we only evaluate them on the smaller datasets. Details about the parameter settings can be found in the appendix in the supplementary material.

## 6.2 EXPERIMENTAL RESULTS

We now present our experimental results. The algorithms are compared in terms of: (i) The function value $f$ of their solution; (ii) The total number of noisy samples of the marginal gain; (iii) The average number of samples per marginal gain estimation (*average samples=total samples/# of evaluated marginal gains*).

Our results for different values of $\epsilon$ and $\kappa$ are presented in Figure 3. From Figures 3(a), 3(c), 3(e) and 3(g), one can see that the total samples required by `CTG` tends to be smaller than those required by `EPS-AP`, `EXP-GREEDY` and `EXP-GREEDY-K`, which demonstrates the advantage of `CTG` in sample efficiency, which was the main goal of the paper. However, on the delicious_300 dataset (Figures 3(b) and 3(d)), the average samples of `EXP-GREEDY-K` is slightly better than `CTG`, and on the other hand `CTG` has significantly better average samples compared to `EXP-GREEDY-K` on the corel_60 dataset (Figures 3(f) and 3(h)). This demonstrates the incomparability of the instance-dependent sample query bounds given for marginal gain computations on `CTG` vs that of `ExpGreedy`.

From the results where we vary $\epsilon$, it can be seen that both the total samples and average samples of our algorithm `CTG` increase less compared with `EPS-AP` and `EXP-GREEDY` as $\epsilon$ decreases (Figures 3(a), 3(b), 3(e) and 3(f)), which corresponds to our theoretical results (see the discussion in Section

H.2 in the appendix). For the experiments comparing different $\kappa$, we can see that the total queries of the EXP-GREEDY and EXP-GREEDY-K increases faster compared with EPS-AP and CTG (Figure 3(c)), which can be attributed to the better dependence on $\kappa$ that TG exhibits compared to the standard greedy algorithm. A result that is a little different from the above is that the number of total queries of EXP-GREEDY-K decreases on dataset corel_60 when $\kappa$ becomes large (Figure 3(g)), which is because when $\kappa$ increases, EXP-GREEDY-K is able to better deal with tiny differences in marginal gains (see the appendix).

Finally, the results on the larger dataset (corel and delicious) of CTG and EPS-AP are presented in Figures 3(i), 3(j), 3(k) and 3(l). Notably, our proposed algorithm (CTG) showcases considerable advantages over the EPS-AP algorithm in terms of both required total samples and average samples.

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

# Appendix

## A ADDITIONAL RELATED WORK

Approximation algorithms for submodular maximization problems with exact value oracle have been extensively studied in the literature Nemhauser et al. (1978); Badanidiyuru & Vondrák (2014); Mirzasoleiman et al. (2015); Balkanski et al. (2019a). For MSMC, the standard greedy algorithm produces a solution set with the best possible $1 - 1/e$ approximation guarantee in $O(n^2)$ queries of $f$. Badanidiyuru & Vondrák (2014) proposed a faster greedy-like algorithm that gives an approximation guarantee of $1 - 1/e - O(\epsilon)$ while reducing the sample complexity to $O(\frac{n}{\epsilon} \log \frac{n}{\epsilon})$.

Another variant is USM Buchbinder et al. (2015); Feige et al. (2011); Buchbinder & Feldman (2018). Notably, Buchbinder et al. (2015) introduced a deterministic algorithm that gives a $1/3$ guarantee in $O(n)$ queries to an oracle for $f$, and a randomized version of their algorithm yields the best possible $1/2$ guarantee in expectation in the same number of queries.

The final variant of submodular maximization we consider is MSMM Balkanski et al. (2019b); Friedrich & Neumann (2014); Fisher et al. (1978). The greedy algorithm only yields an approximation ratio of $1/2$ in this setting Fisher et al. (1978). But by extending the discrete submodular function to its continuous counterpart, known as the multilinear extension (see the definition in Section 2), and by solving the problem in this regime, it is proved that an approximation ratio arbitrarily close to the best possible $1 - 1/e$ can be achieved Badanidiyuru & Vondrák (2014); Calinescu et al. (2011).

Our work is also related to the best-arm-identification in multi-armed bandit literature Audibert et al. (2010); Kaufmann et al. (2016); Jun et al. (2016), where the objective is to estimate the best action by choosing arms and receiving stochastic rewards from the environment. The most widely considered setting is the PAC learning setting Even-Dar et al. (2002); Kalyanakrishnan et al. (2012); Zhou et al. (2014).

Our paper studies the same noisy setting as Singla et al. (2016). There are essentially two versions of `ExpGreedy`, one gives an approximation guarantee of about $1 - 1/e$ with high probability (like our algorithm `CTG` does), and the other gives the same approximation guarantee but is randomized. The benefit of the latter over the former is better sample complexity. The bounds given on the sample complexity of `ExpGreedy` and the ones given in this paper for `CTG` are instance-dependent and incomparable to one another. We discuss how our algorithm relates to `ExpGreedy` in more depth in Section B, but we briefly list here the potential advantages of our algorithm `CTG` compared to `ExpGreedy`: (i) Our algorithm has an approximation guarantee of about $1-1/e$ with high probability as opposed to an approximation guarantee of about $1 - 1/e$ in expectation as in the randomized version of `ExpGreedy`; (ii) Our algorithm is not as sensitive to small differences in marginal gain between elements since it is not based on the standard greedy algorithm as `ExpGreedy` is; (iii) The algorithm of `ExpGreedy` has greater time complexity beyond just the sample complexity because it requires $O(n \log n)$ computations per each noisy query to $\Delta f$; (iv) Our algorithm makes less estimations of $\Delta f$ overall since it is based on a faster variant of the greedy algorithm (`TG`). We further compare the algorithms experimentally in Section 6.2.

### A.1 OTHER NOISY MODEL

If the noisy model is that the the samples are taken from distribution $\mathcal{D}(X)$ to evaluate $f(X)$ instead of the marginal gain, the model also satisfies our setting. This is because if the noisy evaluation of $f(X)$ is R-sub-Gaussian, the noisy evaluation of the marginal gain $\Delta f(X, u)$ can be obtained by taking two noisy samples of $f$ and calculating $\mathcal{D}(X \cup \{u\}) - \mathcal{D}(X)$ and that the difference of two independent sub-Gaussian random variables is also sub-Gaussian.

## B COMPARISON WITH EXPGREEDY

In this section, we provide more discussion about the related algorithm `ExpGreedy` of Singla et al. (2016). `ExpGreedy` combines the standard greedy algorithm with the best arm identification algorithm used in combinatorial bandit literature Chen et al. (2014).

In particular, the standard greedy algorithm for MSMC Nemhauser et al. (1978) goes as follows: A solution $S$ is built by iteratively choosing the element $u \in U$ that maximizes the marginal gain $\Delta f(S, u)$ until the cardinality constraint $\kappa$ is exhausted. `ExpGreedy` follows a setting like ours, so instead of choosing the element of maximum marginal gain at each iteration, they follow the standard greedy algorithm but adaptive sampling following techniques from the best-arm identification problem is done in order to identify the element(s) with the highest marginal gain. The simplest version of their algorithm identifies one element with the highest marginal gain at each iteration, and this version has a guarantee of about $1 - 1/e$ with high probability as in `CTG`. This algorithm is `EXP-GREEDY` in Section 6. However, a downside of this approach is that many samples are often needed to distinguish between elements of nearly the same marginal gain. In contrast, notice that our algorithm `CTG` does not need to compare marginal gains between elements and therefore does not have this issue.

In order to deal with the sample inefficiency, `ExpGreedy` is generalized to a randomized version. The randomized version of `ExpGreedy` involves a subroutine called `TOPX`, which adaptively samples marginal gains until a subset of elements with relatively high marginal gains have been identified. Then a randomly selected element among the subset is added to the solution set. In particular, given an integer $0 < \kappa' \le \kappa$, the `TOPX` algorithm runs TOP-$l$ selection algorithms for each $l \in \{1, 2, ..., \kappa'\}$, and each of the TOP-$l$ selection algorithm runs until it returns a subset of $l$ items with highest marginal gain with high probability. The `TOPX` algorithm stops once there exists some $l$ such that the TOP-$l$ selection algorithm ends. This randomized version of `ExpGreedy` has an almost $1 - 1/e$ approximation guarantee, but it holds in expectation and with high probability. The case where $\kappa' = \kappa$ is `EXP-GREEDY-K` in Section 6.

Now that we have described the two versions of `ExpGreedy` and their corresponding approximation guarantee, we look into more detail about the efficiency of `ExpGreedy` in terms of runtime and sample complexity.

It is proven by Singla et al. (2016) that the number of samples taken for each iteration where an element is added to the solution is at most

$$O\left( n\kappa' R^2 \min\left\{ \frac{4}{\Delta_{\max}^2}, \frac{1}{\epsilon^2} \right\} \log\left( \frac{R^2 \kappa n \min\{\frac{4}{\Delta_{\max}^2}, \frac{1}{\epsilon^2}\}}{\delta} \right) \right)$$

where $\Delta_{\max}$ is the largest difference amongst the first $\kappa'$ element's marginal gains. In other words, this is the number of samples taken each time TOPX is called. Since an element being added involves approximating the marginal gains over all of the elements of $U$, the average sample complexity to compute an approximate marginal gain for a single element is then

$$O\left( \kappa' R^2 \min\left\{ \frac{4}{\Delta_{\max}^2}, \frac{1}{\epsilon^2} \right\} \log\left( \frac{R^2 \kappa n \min\{\frac{4}{\Delta_{\max}^2}, \frac{1}{\epsilon^2}\}}{\delta} \right) \right).$$

We compare the above to a single call of `CS` in our algorithm `CTG`, which is the analogous computation where we are approximating the marginal gain for an element of $U$. Recall from Theorem 3 that the bound for the sample complexity for `CS` is the minimum between

$$\left\{ \frac{2R^2}{\phi^2(S, u)} \log\left( \frac{4R^2 \sqrt{\frac{3nh(\alpha)}{\delta}}}{\phi^2(S, u)} \right), \frac{R^2}{2\epsilon^2} \log\left( \frac{6nh(\alpha)}{\delta} \right) \right\}.$$

If $k' = 1$, i.e. the non-randomized version of `ExpGreedy` that has a similar approximation guarantee to our algorithm `CTG`, then $\Delta_{\max}$ is the difference between the top two marginal gains, which could be very small and therefore the sample complexity quite high. On the other hand, `CS` is not sensitive to this property. In order to make $\Delta_{\max}$ bigger, one could increase $k'$ and use the randomized version of `ExpGreedy`. But this case could have worse sample complexity compared to ours as well. If $\Delta_{\max}$ is small and satisfies that $\Delta_{\max} = O(\epsilon)$, then the sample complexity of `ExpGreedy` is worse than our averaged sample complexity by a factor of at least $O(\kappa')$.

Further, since `ExpGreedy` follows the standard greedy algorithm, there are $\kappa$ calls made to TOPX. In contrast, `CTG` is based on the faster variant of the greedy algorithm, `TG`, and so only requires $O(\log(\kappa))$ iterations over $U$.

Another factor that makes `CTG` preferable to `ExpGreedy` is its run time besides sample complexity. From the description of `ExpGreedy` in Singla et al. (2016), we can see that at each time a noisy

query to $\Delta f$ is taken, the TOP-$l$ selection algorithm updates the confidence interval for all the elements, and then the algorithm sorts all elements to find the set $M_t$ of $l$ elements with highest empirical marginal gain. Then another estimate of the marginal gains is computed to be the empirical mean plus a confidence interval or minus the confidence interval depending on whether the elements are within $M_t$. Next, the algorithm sorts the newly obtained estimates to find the top-$l$ set with respect to the new estimates. However, both `CTG` and `EPS-AP` have more efficient runtime complexity and require only one update of the confidence interval in Line 4 and two comparisons in Line 5 and 7 in `CS`, which is only $O(1)$ in computation.

## C  APPENDIX FOR SECTION 3

In this section, we present the omitted content of Section 3. In Section C.1, we present a comparison of our result with the fixed $\epsilon$-approximation. In Section C.2, we present the proof of Theorem 1. In Section C.3, we present the proof of Theorem 2.

### C.1  COMPARISON OF `CS` TO FIXED $\epsilon$-APPROXIMATION

In this section, we present a comparison of our result with the fixed $\epsilon$-approximation. A fixed $\epsilon$-approximation is essentially when one applies a concentration inequality such as Hoeffding's or the Chernoff Bound for a fixed number of noisy samples such that the empirical mean of the evaluated random variable $X$, which is denoted as $\hat{X}$, satisfies that $|\hat{X} - \mathbb{E}[X]| \leq \epsilon$. (see also discussion in Section 2).

The fundamental reason this approach is less efficient compared to `CS` is that we are only interested in determining whether $f(X)$ is approximately above a threshold or not, not in obtaining a precise approximation. In other words, we don't need the guarantee that the $|\hat{X} - \boldsymbol{E}X| \leq \epsilon$ in Hoeffding's inequality; instead, we care about whether $\boldsymbol{E}X \geq w$. Ideally, we would approximate $f(X)$ just finely enough to determine if it's above the threshold or not. However, this isn't feasible with the fixed $\epsilon$-approximation, because we don't have any prior knowledge of how far $f(X)$ is from the threshold. Consequently, we can't determine the required number of samples, and the fixed $\epsilon$-approximation approach requires that there be a single batch of i.i.d. samples, which limits flexibility.

In contrast, `CS` uses an adaptive sampling approach where samples are iteratively taken one-by-one until an evolving confidence interval crosses a threshold. The goal of `CS` is to use fewer samples compared to a fixed $\epsilon$-approximation. While `CS` might initially seem similar to fixed $\epsilon$-approximation, there are several critical differences that introduce unique technical challenges in its development and analysis:

- Fixed $\epsilon$-approximation approaches have a batch of samples in which a single application of a concentration inequality is applied in order to approximate $\boldsymbol{E}X$. In contrast, in `CS`, we apply a concentration inequality after every single sample, and then take a union bound over all the applications. However, this is challenging because we don't know how many samples we will end up taking to approximate the mean value sufficiently well since that depends on the result of the sampling. So we have to carefully design our confidence intervals.

- Fixed $\epsilon$-approximation approach takes a predetermined number of samples, independent of the sampling results. In contrast, the CS algorithm dynamically determines the number of samples based on the outcomes of previous samples. Additionally, CS reuses samples across multiple applications of concentration bounds, enhancing its efficiency.

- In `CS`, the size of the confidence interval evolves with each additional sample, shrinking as the number of samples increases (see Theorem 1). Additionally, when applying concentration inequalities, the failure probability is adjusted dynamically based on how many samples we've taken so far (see proof of Lemma 6). The benefit of the varying failure probability is that the obtained sample complexity $\frac{8R^2}{\phi_X^2} \log\left(\frac{16R^2}{\phi_X^2}\sqrt{\frac{2}{\delta}}\right)$ won't suffer from small values of $\epsilon$.

- In Theorem 2 and 4, we use a combination of Hoeffding and Chernoff that is well-suited to the threshold algorithms, rather than using one or the other. This approach improves the sample complexity from $O(R^2)$ in Theorem 1 to $O(R)$ when $R$ is large.

CS is in fact related to adaptive approaches used in the Upper Confidence Bound (UCB) algorithm in multi-armed bandit, and is distinct from most existing approaches in submodular optimization, with the notable exception of Singla et al. (2016), which integrates a best-arm identification algorithm into the standard greedy framework.

## C.2 ADDITIONAL LEMMAS AND ANALYSIS OF THEOREM 1

In this section, we present the proof of Theorem 2, which provides the theoretical results of sample complexity and approximation guarantee of the CS algorithm. First of all, we provide the statement of Theorem 1 again.

**Theorem 1.** *For any random variable $X$ that is $R$-sub-Gaussian, if we define $N_1 = 2R^2/\epsilon^2 \log \frac{4}{\delta}$, and $C_t = R\sqrt{\frac{2}{t} \log \frac{8t^2}{\delta}}$, then the algorithm* Confident Sample *achieves that with probability at least $1 - \delta$*

*1.* CS *on input $(w, \epsilon, \delta, \mathcal{D}_X, R)$ takes at most the minimum between*

$$\left\{ \frac{2R^2}{\epsilon^2} \log\left(\frac{4}{\delta}\right), \frac{8R^2}{\phi_X^2} \log\left(\frac{16R^2}{\phi_X^2}\sqrt{\frac{2}{\delta}}\right) \right\}$$

*noisy samples, where $R$ is as defined in Section 2, $\phi_X = \frac{\epsilon + |w - \mathbb{E}X|}{2}$.*

*2. If* CS *returns true, then $\boldsymbol{E}X \geq w - \epsilon$. If* CS *returns false, then $\boldsymbol{E}X \leq w + \epsilon$.*

Before we present the detailed proof, here we provide an overview of the proof. In order for CS to correctly determine whether $\boldsymbol{E}X$ is approximately above or below the threshold $w$, i.e. the second result of Theorem 1, two random events must occur during CS. The first event is that at all iterations during the for loop, the confidence regions around the sample mean ($\hat{X}_t$) contain the true expected value ($\boldsymbol{E}X$). The second event is that after $N_1$ samples taken by the for loop on Line 2, we have achieved an $\epsilon$-additive approximation of the expected value. Basically these two events together mean that CS is correct about the region where $\boldsymbol{E}X$ is throughout the algorithm, and therefore it returns the correct answer to whether $\boldsymbol{E}X$ is approximately above or below the threshold $w$. The following Lemma states that on a run of CS, the two events hold with probability at least $1 - \delta$.

**Lemma 6.** *With probability at least $1 - \delta$, the following two events hold.*

*1. At any time $t \in \mathbb{N}_+$, the sample mean $\widehat{X}_t$ satisfies that $|\widehat{X}_t - \boldsymbol{E}X| \leq C_t$, where $C_t := R\sqrt{\frac{2}{t} \log \frac{8t^2}{\delta}}$.*

*2. The sample mean $\widehat{X}_{N_1}$ at time $N_1 := \frac{2R^2}{\epsilon^2} \log \frac{4}{\delta}$ satisfies that $|\widehat{X}_{N_1} - \boldsymbol{E}X| \leq \epsilon$.*

*Proof.* First, we apply the Hoeffding's inequality on $\widehat{X}_{N_1}$ and it follows that

$$P\left(|\widehat{X}_{N_1} - \boldsymbol{E}X| \geq \epsilon\right) \leq 2\exp\left(-\frac{N_1\epsilon^2}{2R^2}\right) \leq \frac{\delta}{2}.$$

Next, by applying the Hoeffding's inequality for any fixed time $t$, we have that

$$P\left(|\widehat{X}_t - \boldsymbol{E}X| \geq C_t\right) \leq \frac{\delta}{4t^2}.$$

By taking the union bound for any time $t$, it follows that

$$P(\exists t \text{ s.t. } |\widehat{X}_t - \boldsymbol{E}X| \geq C_t)$$

$$\leq \sum_{t=1}^{\infty} P(|\widehat{X}_t - \boldsymbol{E}X| \geq C_t)$$

$$\leq \frac{\delta}{4} \sum_{t=1}^{\infty} \frac{1}{t^2} \leq \frac{\delta}{2}.$$

By taking the union bound again on the two events above, we have that

$$P(|\widehat{X}_{N_1} - \boldsymbol{E}X| \geq \epsilon \text{ or } \exists t \text{ s.t. } |\widehat{X}_t - \boldsymbol{E}X| \geq C_t)$$
$$\leq P\left(|\widehat{X}_{N_1} - \boldsymbol{E}X| \geq \epsilon\right) + P(\exists t \text{ s.t. } |\widehat{X}_t - \boldsymbol{E}X| \geq C_t)$$
$$\leq \delta.$$

$\square$

The second lemma required for establishing Theorem 1 concerns the number of samples that CS takes before its approximation of $\boldsymbol{E}X$ is sufficiently accurate so that it can terminate. The number of samples depends on how far away the true value of $f$ is from the threshold. In particular, Lemma 7 below states that once the confidence interval goes beneath the corresponding $\phi$ value (as defined in Theorem 1), then CS will complete. Lemma 7 and its proof are stated below.

**Lemma 7.** *With probability at least* $1 - \delta$*, when the confidence interval* $C_t$ *satisfies that*

$$C_t \leq \phi_X,$$

*the sampling of X finishes, where* $\phi_X = \frac{\epsilon + |w - \boldsymbol{E}X|}{2}$*.*

*Proof.* If $C_t \leq \frac{\epsilon + w - \boldsymbol{E}X}{2}$, then we have $\boldsymbol{E}X \leq w + \epsilon - 2C_t$. From Lemma 6, we have that with probability at least $1 - \delta$, it holds that $\widehat{X}_t - \boldsymbol{E}X \leq C_t$. Therefore,

$$\widehat{X}_t + C_t$$
$$\leq (\widehat{X}_t - \boldsymbol{E}X) + \boldsymbol{E}X + C_t$$
$$\leq w + \epsilon.$$

Thus the algorithm ends.

Similarly, we consider the case where $C_t \leq \frac{\epsilon - w + \boldsymbol{E}X}{2}$. In this case, we have that $\boldsymbol{E}X \geq 2C_t + w - \epsilon$. Notice that conditioned on the clean event defined in Lemma 6, we have that $\widehat{X}_t - \boldsymbol{E}X \geq -C_t$. Then

$$\widehat{X}_t - C_t \geq \widehat{X}_t - \boldsymbol{E}X$$
$$+ \boldsymbol{E}X - C_t$$
$$\geq -C_t + 2C_t$$
$$+ w - \epsilon - C_t$$
$$= w - \epsilon.$$

Therefore, the algorithm ends. $\square$

Now we present the proof of Theorem 1.

*Proof.* We first prove the result on sample complexity, which is the first result in Theorem 1. From Lemma 7, we have if

$$C_t \leq \phi_X, \tag{1}$$

then the Algorithm 1 finishes. Since $C_t = R\sqrt{\frac{2}{t}\log\frac{8t^2}{\delta}}$, we have the above inequality (1) is equivalent to that

$$\frac{4\log(\sqrt{\frac{8}{\delta}}t)}{t} \leq \frac{\phi_X^2}{R^2}.$$

Since $\sqrt{\frac{8}{\delta}}t \geq 2$, from Lemma 23, we have when

$$t \geq \frac{8R^2}{\phi_X^2}\log(\frac{16R^2}{\phi_X^2}\sqrt{\frac{2}{\delta}}),$$

the above inequality holds and the Algorithm 1 ends. Therefore, the number of samples required is bounded by $\min\{\frac{8R^2}{\phi_X^2}(\log\frac{16R^2}{\phi_X^2}\sqrt{\frac{2}{\delta}}), N_1\}$.

Next, we prove the second result in Theorem 1. If $t = N_1$ when CS ends, then conditioned on the events in Lemma 6, $|\widehat{X}_{N_1} - \boldsymbol{E}X| \leq \epsilon$. Thus if the algorithm returns true, $\boldsymbol{E}X \geq \widehat{X}_t - \epsilon \geq w - \epsilon$. If the output of the algorithm is false, then $\widehat{X}_t \leq w$. Similarly we have that $\boldsymbol{E}X \leq \widehat{X}_t + \epsilon \leq w + \epsilon$. Secondly, let us consider the case where $t < N_1$ when the algorithm CS ends. Conditioned on the second event in Lemma 6, we have if the algorithm CS returns true, $\boldsymbol{E}X \geq \widehat{X}_t - C_t \geq w - \epsilon$. If the output is false, $\boldsymbol{E}X \leq \widehat{X}_t + C_t \leq w + \epsilon$. $\qquad\square$

### C.3 PROOF AND ANALYSIS OF THEOREM 2

In this section, we present the omitted proofs of Theorem 2 in Section 3. Theorem 2 provides another result of the approximation error for the CS algorithm by defining the confidence interval $C_t$ to be $C_t = \frac{3R}{t\alpha}\log\left(\frac{8t^2}{\delta}\right)$ and the worst-case sample complexity $N_1$ to be $N_1 = \frac{3R}{\epsilon\alpha}\log\left(\frac{4}{\delta}\right)$. We begin by stating Theorem 2, followed by the proof of the theorem. Finally, we establish the lemmas crucial to the proof of the theorem.

**Theorem 2.** *For any random variable $X$ that is bounded in the range of $[0, R]$, if we define $C_t = \frac{3R}{t\alpha}\log(\frac{8t^2}{\delta})$, and $N_1 = \frac{3R}{\epsilon\alpha}\log\left(\frac{4}{\delta}\right)$ where $\alpha$ is an additional parameter that controls the multiplicative error rate, the algorithm* Confident Sample *achieves that with probability at least $1 - \delta$, the algorithm* Confident Sample *achieves that with probability at least $1 - \delta$*

1. CS *on input $(w, \epsilon, \delta, \mathcal{D}_X, R)$ takes at most the minimum between*

$$\left\{\frac{3R}{\epsilon\alpha}\log\left(\frac{4}{\delta}\right), \frac{12R}{\alpha\phi'_X}\log\left(\frac{12R}{\alpha\phi'_X}\sqrt{\frac{8}{\delta}}\right)\right\}$$

*noisy samples, $\phi'_X = \frac{\epsilon - \alpha\boldsymbol{E}X + |w - \mathbb{E}X|}{2}$.*

2. *If the output is true, then $(1+\alpha)\boldsymbol{E}X \geq w - \epsilon$. If the output is false, then $(1-\alpha)\boldsymbol{E}X \leq w + \epsilon$.*

*Proof.* First of all, we prove the result on the sample complexity as presented in the first result in Theorem 2. From Lemma 9, we have if

$$C_t \leq \phi'_X,$$

the algorithm ends. By definition of $C_t$, we have that the above result is equivalent to that

$$\frac{3R}{t\alpha}\log(\frac{8t^2}{\delta}) \leq \phi'_X.$$

From Lemma 23, we have that when

$$t \geq \frac{12R}{\alpha\phi'_X}\log\left(\frac{12R}{\alpha\phi'_X}\sqrt{\frac{8}{\delta}}\right)$$

the above inequality holds and thus the algorithm ends. From the description of the algorithm, we have that the number of samples is also bounded by $N_1$. Therefore, the first result in Theorem 2 is proved.

Next, we prove the second result on the difference of $\boldsymbol{E}X$ and $w$. If $t = N_1$ when CS ends, then if the algorithm returns true, we have that with probability at least $1 - \delta$,

$$(1 + \alpha)\mathbb{E}X + \epsilon \geq \widehat{X}_{N_1} \geq w.$$

where the first inequality follows from Lemma 8. If the algorithm returns false and $t = N_1$ when the algorithm ends, then with probability at least $1 - \delta$,

$$(1 - \alpha)\mathbb{E}X - \epsilon \leq \widehat{X}_{N_1} \leq w.$$

Next, we consider the case where $t < N_1$ when the algorithm ends. Conditioned on the first event in Lemma 8 and from the stopping condition of CS, we can see if CS returns true, then

$$(1 + \alpha)\mathbb{E}X + \epsilon \geq \widehat{X}_t - C_t + \epsilon \geq w.$$

If `CS` returns false, then

$$(1 - \alpha)\mathbb{E}X - \epsilon \leq \widehat{X}_t + C_t - \epsilon \leq w.$$

$\square$

We now present the statement and the proofs of the lemmas used in the proof of Theorem 2. We start by introducing Lemma 8, which defines two "clean events".

**Lemma 8.** *With probability at least $1 - \delta$, the following two events hold.*

1. *At any time $t \in \mathbb{N}_+$, the sample average $\widehat{X}_t$ satisfies that $|\widehat{X}_t - \boldsymbol{E}X| \leq \alpha\mathbb{E}X + C_t$, where $C_t := \frac{3R}{t\alpha}\log(\frac{8t^2}{\delta})$.*

2. *The sample average $\widehat{X}_{N_1}$ at time $N_1 := \frac{3R}{\epsilon\alpha}\log\left(\frac{4}{\delta}\right)$ satisfies that $|\widehat{X}_{N_1} - \boldsymbol{E}X| \leq \alpha\boldsymbol{E}X + \epsilon$.*

*Proof.* By applying the Lemma 20, we have that for any fixed time step $t$,

$$P\big(|\widehat{X}_t - \boldsymbol{E}X| > \alpha\mathbb{E}X + C_t\big) \leq 2\exp\{-\frac{t\alpha C_t}{3R}\}$$

$$\leq \frac{\delta}{4t^2}.$$

By taking the union bound over all time step $t \in \mathbb{N}_+$, we have

$$P\big(|\widehat{X}_t - \boldsymbol{E}X| > \alpha\mathbb{E}X + C_t, \forall t\big)$$

$$\leq \sum_{t=1}^{\infty} P\big(|\widehat{X}_t - \boldsymbol{E}X| > \alpha\mathbb{E}X + C_t\big)$$

$$\leq \sum_{t=1}^{\infty} \frac{\delta}{4t^2} \leq \frac{\delta}{2}.$$

Therefore the first event in the lemma holds with probability at least $1 - \delta/2$. By applying the Lemma 20 again, we have that for $t = N_1$,

$$P\big(|\widehat{X}_{N_1} - \boldsymbol{E}X| > \alpha\mathbb{E}X + \epsilon\big) \leq 2\exp\{-\frac{N_1\alpha\epsilon}{3R}\} = \delta/2.$$

It follows that the second event in the lemma holds with probability at least $1 - \delta/2$. By combining the two results and applying the union bound again, we know that with probability at least $1 - \delta$, the two events both hold. $\square$

Next, we prove another lemma that is used in the proof of the sample complexity result in Theorem 2.

**Lemma 9.** *With probability at least $1 - \delta$, when the confidence interval $C_t$ satisfies that*

$$C_t \leq \phi'_X,$$

*the sampling of $X$ finishes, where $\phi'_X = \frac{\epsilon - \alpha\boldsymbol{E}X + |w - \boldsymbol{E}X|}{2}$.*

*Proof.* To prove the lemma, it is equivalent to prove that when $C_t \leq \frac{\epsilon - \alpha\boldsymbol{E}X + w - \boldsymbol{E}X}{2}$ or $C_t \leq \frac{\epsilon - \alpha\boldsymbol{E}X - w + \boldsymbol{E}X}{2}$, the algorithm ends. First of all, if $C_t \leq \frac{\epsilon - \alpha\boldsymbol{E}X + w - \boldsymbol{E}X}{2}$, then $(1 + \alpha)\boldsymbol{E}X + 2C_t \leq w + \epsilon$. Conditioned on the events in Lemma 8, we have that with probability at least $1 - \delta$, it follows that

$$\widehat{X}_t + C_t \leq (1 + \alpha)\boldsymbol{E}X + 2C_t \leq w + \epsilon.$$

Thus the sampling of $X$ ends. Next, if $C_t \leq \frac{\epsilon - \alpha\boldsymbol{E}X - w + \boldsymbol{E}X}{2}$, then $(1 - \alpha)\boldsymbol{E}X - 2C_t \geq w - \epsilon$. By Lemma 8,

$$\widehat{X}_t - C_t \geq (1 - \alpha)\boldsymbol{E}X - 2C_t \geq w - \epsilon.$$

Then the algorithm ends. $\square$

# D   APPENDIX FOR SECTION 4

In this section, we present the omitted content in Section 4, which is organized as follows: In Section D.1, we discuss and compare the theoretical performance of our algorithm, `CTG2`, with the sampling-before-hand algorithm in the context of the influence maximization problem. Next, we provide the proof of our main result, Theorem 3, in Section D.2. Theorem 3 gives the theoretical guarantee of the `CTG` algorithm. Finally, in Section D.3, we provide the brief description of `CTG2` algorithm and the detailed proof of Theorem 4.

## D.1   COMPARING TO SAMPLING-BEFORE-HAND ALGORITHM

Before we describe the sampling-before-hand algorithm and dive into the comparison of this algorithm and `CTG2`, first we present a detailed description of the application of influence maximization. In the influence maximization problem in large-scale networks, the submodular objective is defined as follows:

**Influence aximization**   Suppose the social graph is described by $G = (V, E, \bar{\mathbf{w}})$, where $V$ is the set of nodes with $|V| = n$, $E$ denotes the set of edges, and $\bar{\mathbf{w}}$ is the weight vector defined on the set of edges $E$. Given a seed set $S$, let us define $f(S; \mathbf{w})$ to be the number of nodes reachable from the seed set $S$ under the graph realizations determined by a random weight vector $\mathbf{w}$. Therefore, $f(S; \mathbf{w})$ is bounded by the number of nodes in the graph, i.e., $0 \le f(S; \mathbf{w}) \le n$. The submodular objective is defined as $f(S) = \mathbf{E}_{\mathbf{w} \sim \mathcal{D}(\bar{\mathbf{w}})} f(S; \mathbf{w})$. Here $\mathcal{D}(\bar{\mathbf{w}})$ is the distribution of the weight vector.

The marginal gain can be calculated as

$$\Delta f(S, s) = \mathbf{E}_{\mathbf{w} \sim \mathcal{D}(\bar{\mathbf{w}})} \Delta f(S, s; \mathbf{w})$$
$$= \mathbf{E}_{\mathbf{w} \sim \mathcal{D}(\bar{\mathbf{w}})} f(S \cup \{u\}; \mathbf{w}) - \mathbf{E}_{\mathbf{w} \sim \mathcal{D}(\bar{\mathbf{w}})} f(S),$$

which is also bounded in the range of $[0, n]$.

Next, we describe the sampling-before-hand algorithm, which runs as follows:

1. **Sampling:** The algorithm begins by sampling $N$ i.i.d graph realizations. For the $i$-th graph realization, we denote its weight vector as $\mathbf{w}_i$ and the corresponding function value for a set $S$ as $f_i(S) = f(S; \mathbf{w}_i)$.

2. **Average objective Function:** Next, we define the average function $\hat{f}$ over the sampled graph realizations. This function is given by $\hat{f}(S) = \frac{\sum_{i=1}^{N} f_i(S)}{N}$ for any $S \subseteq U$.

3. **Threshold-greedy algorithm:** We run `Threshold Greedy` (`TG`) with the average function $\hat{f}$ as the submodular objective. The output of the threshold-greedy algorithm is returned as the solution set, denoted as $S$.

### D.1.1   ANALYSIS OF SAMPLING-BEFORE-HAND APPROACH

Now we present the analysis of the sampling-before-hand algorithm. From Lemma 20, and by taking the union bound, we can prove that

$$P(|\hat{f}(X) - f(X)| \ge \alpha f(X) + \epsilon, \forall |X| \le \kappa)$$
$$\le 2n^{\kappa} \exp\{-\frac{N \alpha \epsilon}{3n}\}.$$

Therefore, to guarantee that

$$P(|\hat{f}(X) - f(X)| \ge \alpha f(X) + \epsilon, \forall |X| \le \kappa) \le \delta,$$

it is enough to take

$$N \in \Omega\left(\frac{n}{\alpha \epsilon}\left(\kappa \log n + \log \frac{1}{\delta}\right)\right)$$

number of graph realizations. Since `TG` requires $\frac{n}{\alpha} \log \frac{n}{\alpha}$ number of evaluations of $\hat{f}$. The total number of evaluations of noisy realizations of $f$ would be

$$O\left(\frac{n^2}{\alpha^2 \epsilon} \log \frac{n}{\alpha}\left(\kappa \log n + \log \frac{1}{\delta}\right)\right).$$

Next, we prove the approximation guarantee. From the analysis above, we can see that with probability at least $1 - \delta$

$$
\begin{aligned}
f(S) &\geq \frac{\hat{f}(S) - \epsilon}{1 + \alpha} \\
&\geq (1 - \alpha)\hat{f}(S) - \epsilon \\
&\geq (1 - 1/e - \alpha)(1 - \alpha)\hat{f}(OPT) - \epsilon \\
&\geq (1 - 1/e - 2\alpha)\hat{f}(OPT) - \epsilon \\
&\geq (1 - 1/e - 3\alpha)f(OPT) - 2\epsilon.
\end{aligned}
$$

Now we compare the theoretical guarantees of the sampling-based algorithm and `CTG2`. The theoretical results of `CTG2` are in Theorem 4. Notice that by substituting $\epsilon$ with $\epsilon/k$ in Theorem 4, we obtain a similar approximation guarantee for `CTG2`: $f(S) \geq (1 - 1/e - O(\alpha))f(OPT) - O(\epsilon)$, which matches the result achieved by the sampling-based algorithm.

For the sample complexity, each call of `CS` requires at most the minimum between $O(\frac{\kappa n}{\epsilon\alpha}\log\frac{n}{\delta})$ and $O(\frac{n}{\alpha\phi'(S,u)}\log\frac{n}{\alpha\phi'(S,u)\delta})$ number of samples. The first bound is derived by considering the fixed $\epsilon$- approximation of the marginal gain. If we only consider this bound, then the total number of marginal gains would be $O(\frac{kn^2}{\epsilon\alpha^2}(\log\frac{n}{\alpha})(\log\frac{n}{\delta}))$. In practice, the parameter $\delta$ is usually set to be $O(Poly(1/n))$, such as $O(1/n^2)$. Consequently, the sample complexity of both `CTG2` and the sampling-before-hand approach would be $O(\frac{\kappa n}{\epsilon\alpha}\log n)$. However, it is important to note that `CS` employs the adaptive thresholding technique, which often allows the algorithm to terminate much earlier before reaching the worst-case sample complexity required for fixed-confidence approximation. As a result, `CTG2` can be significantly more sample-efficient in practice.

In comparison to the sampling-before-hand algorithm, `CTG2` offers an additional advantage. The sampling-before-hand algorithm requires obtaining $N$ independent graph realizations and storing all the data at the beginning of the algorithm. However, this can pose practical challenges. Firstly, in scenarios where both $N$ and the graph are exceedingly large, storing all the data might be infeasible. Secondly, in certain applications, such as real-world social networks, obtaining an entire graph realization may not be possible, as we might only be able to sample a portion of the graph at each time.

### D.2 PROOF OF THEOREM 3

In this section, we move towards proving one of our main results, Theorem 3 about `CTG` for the MSMC problem. We state the theorem again as follows.

**Theorem 3.** *Suppose the noisy marginal gain of any subset $S \subseteq U$ and element $s \in U$ is R-sub-Gaussian, then `CTG` makes at most $n\log(\kappa/\alpha)/\alpha$ calls of `CS`. In addition, with probability at least $1 - \delta$, the following statements hold:*

- *The exact function value of the output solution set $S$ satisfies that $f(S) \geq (1 - e^{-1} - \alpha)f(OPT) - 2\kappa\epsilon$;*

- *Each call of `CS` on input $(w, \epsilon, \frac{2\delta}{3nh(\alpha)}, \mathcal{D}(S,u), R)$ takes at most the minimum between*

$$
\frac{8R^2}{\phi^2(S,u)}\log\left(\frac{16R^2\sqrt{\frac{3nh(\alpha)}{\delta}}}{\phi^2(S,u)}\right)
$$

*and*

$$
\frac{2R^2}{\epsilon^2}\log\left(\frac{6nh(\alpha)}{\delta}\right)
$$

*noisy samples. Here $OPT$ is an optimal solution to the MSMC problem, $\phi(S,u) = \frac{\epsilon + |w - \Delta f(S,u)|}{2}$, and $h(\alpha) = \frac{\log(\kappa/\alpha)}{\alpha}$.*

To prove the theorem, we first present a series of needed lemmas. In order for the guarantees of Theorem 3 to hold, two random events must occur during `CTG`. The first event is that the estimate of

the max singleton value of $f$ on Line 4 in `CTG` is an $\epsilon$-approximation of its true value. More formally, we have the following lemma.

**Lemma 10.** *With probability at least $1 - \delta/3$, we have $\max_{s \in U} f(s) - \epsilon \le d \le \max_{s \in U} f(s) + \epsilon$.*

*Proof.* For a fix $s \in U$, by Hoeffding's inequality we would have that

$$P(|\hat{f}(s) - f(s)| \ge \epsilon) \le \frac{\delta}{3n}. \tag{2}$$

Taking a union bound over all elements we would have that

$$P(\exists s \in U, s.t. |\hat{f}(s) - f(s)| \ge \epsilon) \le \frac{\delta}{3}.$$

Then with probability at least $1 - \frac{\delta}{3}$, $|\hat{f}(s) - f(s)| \le \epsilon$ for all $s \in U$. It then follows that $\forall s \in U$, $f(s) - \epsilon \le \hat{f}(s) \le f(s) + \epsilon$. Therefore

$$\max_{s \in U}(f(s) - \epsilon) \le \max_{s \in U} \hat{f}(s) \le \max_{s \in U}(f(s) + \epsilon).$$

Thus we have

$$\max_{s \in U} f(s) - \epsilon \le d \le \max_{s \in U} f(s) + \epsilon.$$

$\square$

The second event is that for all calls of `CS`, the result in Theorem 1 holds, which is stated formally as follows.

**Lemma 11.** *With probability at least $1 - 2\delta/3$, we have that during each call of `CS` with the solution set $S$ and element $u$, the output satisfies that if $thre$ is true, then $\Delta f(S, u) \ge w - \epsilon$. If $thre$ is false, then $\Delta f(S, u) \le w + \epsilon$.*

*Proof.* First, since each sampling result of the marginal gain is assumed to be $R$-sub-Gaussian, by applying the result in Theorem 1, we can prove that for each call of `CS` during `CTG` with a fixed solution set $S$ and evaluated element $u$ as input, and with probability at least $1 - \frac{2\delta}{3nh(\alpha)}$, if the output of `CS` is true, then $\Delta f(S, u) \ge w - \epsilon$. Otherwise, $\Delta f(S, u) \le w + \epsilon$. Since there are $n$ elements in the universe and the number of iterations in Algorithm 2 is bounded by $\frac{\log \kappa/\alpha}{\log(1/(1-\alpha))} \le h(\alpha)$, there are at most $nh(\alpha)$ number of marginal gains to evaluate in Algorithm 2. Therefore, by taking the union bound we have that with probability at least $1 - 2\delta/3$, the statement holds. $\square$

With the above Lemma 10 and Lemma 11, and by taking the union bound, we have that with probability at least $1 - \delta$, the two events both hold during the `CTG`. Our next step is to show that if both of the events occur during `CTG`, the approximation guarantees and sample complexity of Theorem 3 hold. To this end, we need the following Lemma 12.

**Lemma 12.** *Assume the events defined in Lemma 10 and Lemma 11 above hold during `CTG`. Then for any element $s$ that is added to the solution set $S$, the following statement holds.*

$$\Delta f(S, s) \ge \frac{1 - \alpha}{\kappa}(f(OPT) - f(S)) - 2\epsilon.$$

*Proof.* At the first iteration, if an element $s$ is added to the solution set, it holds by Lemma 10 that $\Delta f(S, s) \ge w - \epsilon$. Since at the first iteration $w = d$ and $d \ge \max_{s \in U} f(s) - \epsilon$. It follows that $\Delta f(S, s) \ge \max_{s \in U} f(s) - 2\epsilon$. By submodularity we have that $\kappa \max_{s \in U} f(s) \ge f(OPT)$. Therefore, $\Delta f(S, s) \ge \frac{f(OPT) - f(S)}{\kappa} - 2\epsilon$.

At iteration $i$ where $i > 1$, if an element $o \in OPT$ is not added to the solution set, then it is not added to the solution at the last iteration, where the threshold is $\frac{w}{1-\alpha}$. By Lemma 6, we have

$\Delta f(S, o) \leq \frac{w}{1-\alpha} + \epsilon$. Since for any element $s$ that is added to the solution at iteration $i$, by Lemma 6 it holds that $\Delta f(S, s) \geq w - \epsilon$. Therefore, we have

$$\Delta f(S, s) \geq w - \epsilon$$
$$\geq (1-\alpha)(\Delta f(S, o) - \epsilon) - \epsilon$$
$$\geq (1-\alpha)\Delta f(S, o) - 2\epsilon.$$

By submodularity, it holds that $\Delta f(S, s) \geq (1-\alpha)\frac{f(OPT) - f(S)}{\kappa} - 2\epsilon$. $\qquad\square$

We now prove the main result, Theorem 3, which relies on the previous Lemma 10, 11 and 12.

*Proof.* The events defined in Lemma 10, 11 hold with probability at least $1 - \delta$ by combining Lemma 10, 11, and taking the union bound. Therefore in order to prove Theorem 3, we assume that both the two events have occurred. The proof of the first result in the theorem depends on the Lemma 12. First, consider the case where the output solution set satisfies $|S| = \kappa$. Denote the solution set $S$ after the $i$-th element is added as $S_i$. Then by Lemma 12, we have

$$f(S_{i+1}) \geq \frac{1-\alpha}{\kappa} f(OPT) + (1 - \frac{1-\alpha}{\kappa})f(S_i) - 2\epsilon.$$

By induction, we have that

$$f(S_\kappa) \geq (1 - (1 - \frac{1-\alpha}{\kappa})^k)\{f(OPT) - \frac{2\kappa\epsilon}{1-\alpha}\}$$
$$\geq (1 - e^{-1+\alpha})\{f(OPT) - \frac{2\kappa\epsilon}{1-\alpha}\}$$
$$\geq (1 - e^{-1} - \alpha)\{f(OPT) - \frac{2\kappa\epsilon}{1-\alpha}\}$$
$$\geq (1 - e^{-1} - \alpha)f(OPT) - 2\kappa\epsilon.$$

If the size of the output solution set $S$ is smaller than $\kappa$, then any element $o \in OPT$ that is not added to $S$ at the last iteration satisfies that $\Delta f(S, o) \leq w + \epsilon$. Since the threshold $w$ in the last iteration satisfies that $w \leq \frac{\alpha d}{\kappa}$, we have

$$\Delta f(S, o) \leq \frac{\alpha d}{\kappa} + \epsilon.$$

It follows that

$$\sum_{o \in OPT \setminus S} \Delta f(S, o) \leq \alpha(\max_{s \in S} f(s) + \epsilon) + \kappa\epsilon$$
$$\leq \alpha f(OPT) + 2\kappa\epsilon.$$

By submodularity and monotonicity of $f$, we have $f(S) \geq (1-\alpha)f(OPT) - 2\kappa\epsilon$.

$\qquad\square$

### D.3 ANALYSIS OF CTG2

In this section, we analyze Theorem 4, which establishes the sample complexity and approximation ratio guarantees for the solution obtained by `Confident Threshold Greedy2` (CTG2). CTG2 is an algorithm for the MSMC problem where only noisy queries to $\Delta f$ are available. The corresponding algorithm description is presented in Algorithm 3.

First of all, we give a brief description of the CTG2 algorithm. CTG2 shares a similar idea with the CTG algorithm presented in Section 4. Both of the two algorithms utilize CS to determine if the expectation of the evaluated marginal gain is approximately above a threshold $w$. However, they differ in their error approximation guarantees on the expectation of evaluated marginal gain. Specifically, CTG invokes the `Confident Sample` procedure (CS) with the following inputs: threshold $w$, approximation error bound $\epsilon$, error probability $\frac{2\delta}{3nh'(\alpha)}$ where $h'(\alpha) = \frac{3\log(3\kappa/\alpha)}{\alpha}$, random distribution $\mathcal{D}(S, u)$, and upper bound of the noisy marginal gain $R$ as input. Different

---

**Algorithm 3:** `Confident Threshold Greedy2 (CTG2)`

---

1: **Input:** $\epsilon, \delta, \alpha$
2: $N_3 \leftarrow \frac{9R}{\epsilon\alpha} \log \frac{6n}{\delta}$
3: **for all** $s \in U$ **do**
4:    $\hat{f}_{N_3}(s) \leftarrow$ sample mean over $N_3$ samples from $\mathcal{D}(\emptyset, s)$
5: **end for**
6: $d := \max_{s \in U} \hat{f}_{N_3}(s)$,
7: $w \leftarrow d, S \leftarrow \emptyset$
8: **while** $w > \frac{\alpha d}{3\kappa}$ **do**
9:    **for all** $u \in U$ **do**
10:      **if** $|S| < \kappa$ **then**
11:        thre = `Confident Sample` $(w, \epsilon, \frac{2\delta}{3nh'(\alpha)}, \mathcal{D}(S, u), R)$
12:        **if** thre **then**
13:          $S \leftarrow S \cup \{u\}$
14:        **end if**
15:      **end if**
16:    **end for**
17:    $w = w(1 - \alpha/3)$
18: **end while**
19: **return** $S$

---

from the subroutine algorithm `CS` in `CTG`, the worst-case query complexity $N_1$ and confidence interval $C_t$ in `CS` are defined as in Theorem 2 with the multiplicative input parameter set to $\alpha/3$. Therefore, the output of `CS` in `CTG2` satisfies that with high probability, if the output is true, then $(1 + \alpha/3)\Delta f(S, u) \geq w - \epsilon$. If the output is false, then $(1 - \alpha/3)\Delta f(S, u) \leq w + \epsilon$.

Next, we present the analysis of Theorem 4.

**Theorem 4.** *Suppose the noisy marginal gain of any subset $S \subseteq U$ and element $s \in U$ is bounded in $[0, R]$, CTG2 makes at most $3n \log(\kappa/\alpha)/\alpha$ calls of CS. In addition, with probability at least $1 - \delta$, the following statements hold:*

- *The exact function value of the output solution set $S$ satisfies that $f(S) \geq (1 - e^{-1} - \alpha)f(OPT) - 2\kappa\epsilon$;*
- *Each call of CS on input $(w, \epsilon, \frac{2\delta}{3nh'(\alpha)}, \mathcal{D}(S, u), R)$ takes at most the minimum between*

$$\frac{9R}{\epsilon\alpha} \log \left( \frac{6nh'(\alpha)}{\delta} \right)$$

   *and*

$$\frac{36R}{\alpha\phi'(S, u)} \log \left( \frac{36R}{\alpha\phi'(S, u)} \sqrt{\frac{12nh'(\alpha)}{\delta}} \right)$$

   *noisy samples. Here $OPT$ is an optimal solution to the MSMC problem, $\phi'(S, u) = \frac{\epsilon - \alpha\Delta f(S,u)/3 + |w - \Delta f(S,u)|}{2}$, and $h'(\alpha) = \frac{3}{\alpha} \log \left( \frac{3\kappa}{\alpha} \right)$.*

Now we present the proof of Theorem 4. The organization of the proof for Theorem 4 is as follows: we begin by presenting the proof of the Theorem 4. Then the proofs of two lemmas, Lemma 13 and Lemma 14, that are used in the proof of Theorem 4 are presented.

*Proof.* First, since the number of iterations in the while loop from Line 9 to Line 17 in `CTG2` (see Algorithm 3) is upper bounded by $\frac{3}{\alpha} \log \frac{3\kappa}{\alpha}$, `CTG2` makes at most $\frac{3n}{\alpha} \log \frac{3\kappa}{\alpha}$ calls of `CS`. Next, we prove the second result in Theorem 4, which guarantees the upper bound on the required number of samples. By applying Lemma 13 on the sampling of the noisy marginal gain of $\Delta f(S, u)$, we can see that with probability at least $1 - \delta$, for each call of `CS`, we have that the number of noisy queries is bounded by the minimum between $\frac{9R}{\epsilon\alpha} \log \left( \frac{6nh'(\alpha)}{\delta} \right)$ and $\frac{36R}{\alpha\phi'(S,u)} \log \left( \frac{36R}{\alpha\phi'(S,u)} \sqrt{\frac{12nh'(\alpha)}{\delta}} \right)$.

Now we prove the first result. Since the proof of the first result is similar to the proof of Theorem 3, here we provide a proof sketch and omit the details. First of all, by Lemma 14, we have

$$f(S_{i+1}) \geq \frac{1 - \alpha}{\kappa} f(OPT) + (1 - \frac{1 - \alpha}{\kappa}) f(S_i) - 2\epsilon.$$

Let us denote the solution set $S$ after the $i$-th element is added as $S_i$. Notice that the result in Lemma 12 is the same as Lemma 14. Therefore, following the same proof as that in Theorem 3, we would get that if $|S| = \kappa$, then by induction

$$f(S_\kappa) \geq (1 - e^{-1} - \alpha) f(OPT) - 2\kappa\epsilon.$$

If the size of the output solution set $S$ is smaller than $\kappa$, then any element $o \in OPT$ that is not added to $S$ at the last iteration satisfies that $(1 - \alpha/3)\Delta f(S, o) \leq w + \epsilon$. Since at the last iteration $w \leq \frac{\alpha d}{3\kappa}$, and that conditioned on the events in Lemma 13, $d \leq (1 + \alpha/3) \max_{s \in U} f(s) + \epsilon$, it follows that

$$(1 - \alpha/3)\Delta f(S, o) \leq \frac{\alpha}{3\kappa} \{(1 + \alpha/3) \max_{s \in U} f(s) + \epsilon\} + \epsilon$$

By submodularity and monotonicity of $f$, we have

$$\begin{aligned}
f(OPT) - f(S) &\leq \sum_{o \in OPT} \Delta f(S, o) \\
&\leq \frac{\alpha}{3(1 - \alpha/3)} \{(1 + \alpha/3) \max_{s \in U} f(s) + \epsilon\} \\
&\quad + \frac{\kappa\epsilon}{(1 - \alpha/3)} \\
&\leq \alpha \max_{s \in U} f(s) + 2\kappa\epsilon \\
&\leq \alpha f(OPT) + 2\kappa\epsilon.
\end{aligned}$$

Then we have $f(S) \geq (1 - \alpha) f(OPT) - 2\kappa\epsilon$. $\qquad\square$

The proof of the above Theorem 4 depends on Lemma 14. Before proving Lemma 14, we first prove the Lemma 13.

**Lemma 13.** *With probability at least $1 - \delta$, the following two events hold.*

1. *$(1 - \alpha/3) \max_{s \in U} f(s) - \epsilon \leq d \leq (1 + \alpha/3) \max_{s \in U} f(s) + \epsilon$.*

2. *During each call of $CS$ on input $(w, \epsilon, \frac{2\delta}{3nh'(\alpha)}, \mathcal{D}(S, u), R)$, if the output is true, then $(1 + \alpha/3)\Delta f(S, u) \geq w - \epsilon$. If the output is false, then $(1 - \alpha/3)\Delta f(S, u) \leq w + \epsilon$. In addition, the number of samples taken by $CS$ is at most the minimum between*

$$\frac{9R}{\epsilon\alpha} \log\left(\frac{6nh'(\alpha)}{\delta}\right) \tag{3}$$

*and*

$$\frac{36R}{\alpha\phi'(S, u)} \log\left(\frac{36R}{\alpha\phi'(S, u)} \sqrt{\frac{12nh'(\alpha)}{\delta}}\right), \tag{4}$$

*where $\phi'(S, u) = \frac{\epsilon - \alpha\Delta f(S,u)/3 + |w - \Delta f(S,u)|}{2}$, and $h'(\alpha) = \frac{3}{\alpha} \log\left(\frac{3\kappa}{\alpha}\right)$.*

*Proof.* First of all, by applying the inequality in Lemma 20, we have that for fixed element $s \in U$

$$P\left(|\hat{f}_{N_3}(s) - f(s)| \geq \frac{\alpha}{3} f(s) + \epsilon\right) \leq \frac{\delta}{3n}.$$

Taking a union bound over all elements in $U$, it follows that

$$P\left(|\hat{f}_{N_3}(s) - f(s)| \geq \frac{\alpha}{3} f(s) + \epsilon, \forall s \in U\right) \leq \frac{\delta}{3},$$

where $N_3 = \frac{9R}{\epsilon\alpha} \log \frac{6n}{\delta}$. Therefore, with probability at least $1 - \delta/3$, we have $|\hat{f}_{N_3}(s) - f(s)| \leq \frac{\alpha}{3} f(s) + \epsilon$ for each $s \in U$. Denote $s_1 = \arg\max_{s \in U} \hat{f}_{N_3}(s)$ and $s_2 = \arg\max_{s \in U} f(s)$. It follows that with probability at least $1 - \delta/3$, we have that

$$d = \hat{f}_{N_3}(s_1) \leq (1 + \alpha/3)f(s_1) + \epsilon \leq (1 + \alpha/3)f(s_2) + \epsilon,$$

and that

$$d = \hat{f}_{N_3}(s_1) \geq \hat{f}_{N_3}(s_2) \geq (1 - \alpha/3)f(s_2) - \epsilon.$$

Since $d = \max_{s \in U} \hat{f}_{N_3}(s) = \hat{f}_{N_3}(s_1)$ and $f(s_2) = \max_{s \in U} f(s)$, the first result holds with probability at least $1 - \delta/3$.

Next, we prove the second result. For each call of the sampling algorithm CS with fixed input $(w, \epsilon, \frac{2\delta}{3nh'(\alpha)}, \mathcal{D}(S, u), R)$, and given that $N_1$ and $C_t$ are defined in accordance with Theorem 2 with the multiplicative error parameter set to $\alpha/3$, we can leverage the second result in Theorem 2. Consequently, with probability at least $1 - \frac{2\delta}{3nh'(\alpha)}$, the following two things hold:

1. If the output of CS is true, then $(1 + \alpha/3)\Delta f(S, s) \geq w - \epsilon$. If the output is false, then $(1 - \alpha/3)\Delta f(S, s) \leq w + \epsilon$.

2. The number of noisy queries is bounded by the minimum between (3) and (4) in the lemma.

Since there are at most $\frac{\log(3\kappa/\alpha)}{\log \frac{1}{1-\alpha/3}} \leq h'(\alpha) = \frac{3}{\alpha} \log \frac{3\kappa}{\alpha}$ number of iterations in CTG2, there are at most $nh'(\alpha)$ calls of CS. Therefore, by taking the union bound we have that with probability at least $1 - 2\delta/3$, the two events defined above hold for all calls to CS during CTG2. By taking the union bound again, we have that with probability at least $1 - \delta$, the two results in the lemma both hold. $\square$

Now we prove the Lemma 14.

**Lemma 14.** *Assume the events defined in Lemma 13 hold during CTG2. Then for any element $s$ that is added to the solution set $S$, the following statement holds.*

$$\Delta f(S, s) \geq \frac{1 - \alpha}{\kappa}(f(OPT) - f(S)) - 2\epsilon.$$

*Proof.* At the first iteration, if an element $s$ is added to the solution set, it holds by Lemma 13 that $(1 + \frac{\alpha}{3})\Delta f(S, s) \geq w - \epsilon$. Since at the first iteration $w = d$ and $d \geq (1 - \alpha/3)\max_{s \in U} f(s) - \epsilon$. It follows that $\Delta f(S, s) \geq \frac{1-\alpha/3}{1+\alpha/3}\max_{s \in U} f(s) - \frac{2\epsilon}{1+\alpha/3} \geq (1 - \alpha)\max_{s \in U} f(s) - 2\epsilon$. By submodularity we have that $\kappa \max_{s \in U} f(s) \geq f(OPT)$. Therefore, $\Delta f(S, s) \geq \frac{1-\alpha}{\kappa}(f(OPT) - f(S)) - 2\epsilon$.

At iteration $i$ where $i > 1$, if an element $o \in OPT$ is not added to the solution set, then it is not added to the solution set at the last iteration, where the threshold is $\frac{w}{1-\alpha/3}$. By Lemma 13, we have $(1 - \alpha/3)\Delta f(S, o) \leq \frac{w}{1-\alpha/3} + \epsilon$. For any element $s$ that is added to the solution at iteration $i$, by Lemma 13 it holds that $(1 + \alpha/3)\Delta f(S, s) \geq w - \epsilon$. Therefore, we have

$$\Delta f(S, s) \geq \frac{w - \epsilon}{1 + \alpha/3}$$

$$\geq \frac{(1 - \alpha/3)^2 \Delta f(S, o) - (1 - \alpha/3)\epsilon - \epsilon}{1 + \alpha/3}$$

$$\geq (1 - \alpha)\Delta f(S, o) - 2\epsilon.$$

By submodularity, it holds that $\Delta f(S, s) \geq (1 - \alpha)\frac{f(OPT) - f(S)}{\kappa} - 2\epsilon$. $\square$

## E  NON-MONOTONE SUBMODULAR OBJECTIVES

In Section 4 and Section 5, we employ the adaptive sampling algorithm CS as a subroutine in algorithms that share the same intuition as TG to determine if the marginal gain is approximately

above or below the threshold $w$. In this section, we demonstrate that CS can also be employed to develop a deterministic algorithm for the Submodular Maximization (USM) problem, following a similar idea as in Buchbinder et al. (2015). Here we assume that the sampling of the marginal gain $\Delta f(S, s)$ is $R$-sub-Gaussian for any $S \subseteq U$ and $s \in U$.

We propose the algorithm CDG, which is based upon the deterministic algorithm presented in Buchbinder et al. (2015) ("Double Greedy") for USM in the noise-free setting, with our procedure CS integrated into it in order to deal with the noisy access to $f$. Here the parameters $N_1$ and $C_t$ in the subroutine algorithm CS are defined in accordance with Theorem 1. We denote the sets $A$ and $B$ after the $i$-th iteration in CDG as $A_i$ and $B_i$, and the element processed in the $i$-th iteration as $u_i$. Pseudocode for CDG is presented in Algorithm 4.

We start by briefly describing the deterministic algorithm in Buchbinder et al. (2015). In particular, the algorithm of Buchbinder et al. (2015) maintains two sets $A$ and $B$ as it makes a single pass through the ground set $U$ in the order $u_1, ..., u_n$. At each element $u_i$, the algorithm evaluates whether $\Delta f(A_{i-1}, u_i)$, the marginal gain of adding the new element $u_i$, surpasses the loss incurred by removing it from set $B_{i-1}/\{u_i\}$, which is $-\Delta f(B_{i-1}/\{u_i\}, u_i)$. If $\Delta f(A_{i-1}, u_i) \geq -\Delta f(B_{i-1}/\{u_i\}, u_i)$, then $u_i$ is added to the final solution set. Otherwise, it is removed from $B_{i-1}$. Our insight is that this procedure in fact is asking about whether the value of the function $\Delta f(A_{i-1}, u_i) + \Delta f(B_{i-1}/\{u_i\}, u_i)$ is above or below the threshold 0.

It is important to note that CS cannot be used as a subroutine in the randomized algorithm with a $1/2$ approximation guarantee as presented in Buchbinder et al. (2015). This is due to a fundamental difference in the requirements of the two algorithms. The randomized algorithm in Buchbinder et al. (2015) requires knowing the exact ratio of $\frac{\Delta f(A_{i-1}, u_i)}{\Delta f(A_{i-1}, u_i) + \Delta f(B_{i-1}/\{u_i\}, u_i)}$, while CS only guarantees the difference between the mean of a random variable and a threshold value $w$. Therefore, in the deterministic algorithm, we can apply CS to find whether the expectation of $X_i = \widetilde{\Delta f}(A_{i-1}, u_i) + \widetilde{\Delta f}(B_{i-1}/\{u_i\}, u_i)$ is approximately above or below 0.

We now present our theoretical guarantees for CDG below in Theorem 15. The proof of Theorem 15 can be found in the supplementary material. We note that our algorithm CDG achieves nearly the same approximation guarantee as that of Buchbinder et al. (2015), but with a small penalty due to the noisy setting.

**Theorem 15.** *CDG makes $n$ calls of CS. In addition, with probability at least $1 - \delta$, the following statements hold:*

1. *The exact function value of the output solution set $S$ satisfies that $f(S) \geq \frac{f(OPT)}{3} - \epsilon$;*
2. *Each call of CS on input $(0, \frac{3\epsilon}{n}, \frac{\delta}{n}, \mathcal{D}_{X_i}, \sqrt{2}R)$ takes at most the minimum between*

$$\left\{ \frac{4n^2 R^2}{9\epsilon^2} \log\left(\frac{4n}{\delta}\right), \frac{16R^2}{\phi_i^2} \log\left(\frac{32R^2}{\phi_i^2} \sqrt{\frac{2n}{\delta}}\right) \right\}$$

*noisy samples. Here $OPT$ is an optimal solution to the USM problem, and*

$$\phi_i := \frac{3\epsilon/n + |\boldsymbol{E}X_i|}{2}$$
$$= \frac{3\epsilon/n + |\Delta f(A_{i-1}, u_i) + \Delta f(B_{i-1}/\{u_i\}, u_i)|}{2}.$$

From Theorem 15, we can see that CDG achieves an approximation guarantee that is arbitrarily close to $1/3$, which matches the result of the deterministic algorithm in Buchbinder et al. (2015).

Now we start to prove the results in Theorem 15. Notice that conditioned on the solution set $A_{i-1}$ and $B_{i-1}$, the random variables $\widetilde{\Delta f}(A_{i-1}, u_i)$ and $\widetilde{\Delta f}(B_{i-1}/\{u_i\}, u_i)$ are $R$-sub-Gaussian. Therefore, $X_i := \widetilde{\Delta f}(A_{i-1}, u_i) + \widetilde{\Delta f}(B_{i-1}/\{u_i\}, u_i)$ is $\sqrt{2}R$-sub-Gaussian, the second result is implied by applying Theorem 1 immediately. To prove the first result in Theorem 15, we need the following lemma.

---

**Algorithm 4:** `Confident Double Greedy` (CDG )

---

1: **Input:** $\epsilon, \delta$
2: $A \leftarrow \emptyset, B \leftarrow U$
3: **for all** $u \in U$ **do**
4:     Define r.v. $X = \widetilde{\Delta} f(A, u) + \widetilde{\Delta} f(B/\{u\}, u)$,
5:     thre = `Confident Sample` $(0, \frac{3\epsilon}{n}, \frac{\delta}{n}, \mathcal{D}_X, \sqrt{2}R)$
6:     **if** $thre$ **then**
7:        $A \leftarrow A \cup \{u\}$
8:     **else**
9:        $B \leftarrow B/\{u\}$
10:    **end if**
11: **end for**
12: **return** $A$

---

**Lemma 16.** *With probability at least $1 - \frac{\delta}{n}$, the $i$-th call of* CS *satisfies the following inequality*

$$f(A_{i-1} \cup OPT_{i-1}) - f(A_i \cup OPT_i) \leq$$

$$[f(A_i) - f(A_{i-1})] + [f(B_i) - f(B_{i-1})] + \frac{3\epsilon}{n}. \tag{5}$$

*where $OPT_i$ is the set of all elements from $OPT$ that arrives after the $i$-th iteration.*

*Proof.* From the statement of the algorithm, we know that the element $u_i$ is added to the solution if and only if the output of CS is true. By applying the results in Theorem 1, we have that for each fixed $i$, with probability at least $1 - \delta/n$ if $u_i$ is added, then $\Delta f(A_{i-1}, u_i) \geq -\Delta f(B_{i-1}/\{u_i\}, u_i) - \frac{3\epsilon}{n}$. Otherwise, $\Delta f(A_{i-1}, u_i) \leq -\Delta f(B_{i-1}/\{u_i\}, u_i) + \frac{3\epsilon}{n}$. Let us denote the above event as $\mathcal{E}_i$, we discuss the following four cases in our analysis

1. If $u_i \in A_i$, and $u_i \in OPT$, then
$$f(A_{i-1} \cup OPT_{i-1}) - f(A_i \cup OPT_i) = 0$$
Notice that $u_i \in A_i$, then conditioned on $\mathcal{E}_i$, we have $\Delta f(A_{i-1}, u_i) \geq -\Delta f(B_{i-1}/\{u_i\}, u_i) - \frac{3\epsilon}{n}$. By submodularity, $\Delta f(B_{i-1}/\{u_i\}, u_i) \leq \Delta f(A_{i-1}, u_i)$. Then it follows that $\Delta f(A_{i-1}, u_i) + \frac{3\epsilon}{2n} \geq 0$. Therefore, the term on the right-hand side of (5) satisfies
$$[f(A_i) - f(A_{i-1})] + [f(B_i) - f(B_{i-1})] + \frac{3\epsilon}{n}$$
$$= \Delta f(A_{i-1}, u_i) + \frac{3\epsilon}{n} \geq 0.$$

2. If $u_i \in A_i$, and $u_i \notin OPT$, then
$$f(A_{i-1} \cup OPT_{i-1}) - f(A_i \cup OPT_i)$$
$$= -\Delta f(A_{i-1} \cup OPT_i, u_i)$$
$$\leq -\Delta f(B_{i-1}/\{u_i\}, u_i),$$
where the inequality is obtained by submodularity. The right-hand side in (5) is
$$[f(A_i) - f(A_{i-1})] + [f(B_i) - f(B_{i-1})] + \frac{3\epsilon}{n}$$
$$= \Delta f(A_{i-1}, u_i) + \frac{3\epsilon}{n}.$$
Notice that $u_i \in A_i$, then conditioned on $\mathcal{E}_i$, we have $\Delta f(A_{i-1}, u_i) \geq -\Delta f(B_i/\{u_i\}, u_i) - \frac{3\epsilon}{n}$. Therefore,
$$[f(A_i) - f(A_{i-1})] + [f(B_i) - f(B_{i-1})] + \frac{3\epsilon}{n}$$
$$= \Delta f(A_{i-1}, u_i) + \frac{3\epsilon}{n}$$
$$\geq -\Delta f(B_{i-1}/\{u_i\}, u_i).$$

3. If $u_i \notin A_i$, and $u_i \notin OPT$, then

$$f(A_{i-1} \cup OPT_{i-1}) - f(A_i \cup OPT_i) = 0.$$

Similarly as the first case, we have that $-\Delta f(B_{i-1}/\{u_i\}, u_i) \geq \frac{3\epsilon}{2n}$. Since the right-hand side is $-\Delta f(B_{i-1}/\{u_i\}, u_i) + \frac{3\epsilon}{n}$, the inequality holds.

4. If $u_i \notin A_i$, and $u_i \in OPT$, then

$$f(A_{i-1} \cup OPT_{i-1}) - f(A_i \cup OPT_i)$$
$$= \Delta f(A_{i-1} \cup OPT_i, u_i) \leq \Delta f(A_{i-1}, u_i),$$

where the inequality holds by submodularity. Conditioned on the event $\mathcal{E}_i$, it follows that $\Delta f(A_{i-1}, u_i) \leq -\Delta f(B_i/\{u_i\}, u_i) + \frac{3\epsilon}{n}$. Since the right-hand side is

$$[f(A_i) - f(A_{i-1})] + [f(B_i) - f(B_{i-1})] + \frac{3\epsilon}{n}$$
$$= -\Delta f(B_i/\{u_i\}, u_i) + \frac{3\epsilon}{n},$$

the result is proved.

$\square$

Now we prove Theorem 15.

*Proof.* Define the event

$$\mathcal{F}_i = \{f(A_{i-1} \cup OPT_{i-1}) - f(A_i \cup OPT_i) \leq$$
$$[f(A_i) - f(A_{i-1})] + [f(B_i) - f(B_{i-1})] + \frac{3\epsilon}{n}\}.$$

From Lemma 16 and by taking the union bound, it follows that

$$P(\mathcal{F}_i, \forall i \in [n]) \geq 1 - \delta$$

Therefore, with probability at least $1 - \delta$, $\mathcal{F}_i$ holds for all $i$. Then by summing over all $i$, we would get

$$\sum_{i=1}^{n} f(A_{i-1} \cup OPT_{i-1}) - f(A_i \cup OPT_i) \leq$$

$$\sum_{i=1}^{n} \{[f(A_i) - f(A_{i-1})]$$
$$+ [f(B_i) - f(B_{i-1})]\} + 3\epsilon.$$

It follows that

$$f(OPT_0) - f(A_n) \leq$$
$$[f(A_n) - f(A_0)] + [f(B_n) - f(B_0)]\} + 3\epsilon.$$

Since the submodular function is nonnegative, and that $f(A_n) = f(B_n)$, $OPT_0 = OPT$, it follows that $f(A) \geq f(OPT)/3 - \epsilon$.

$\square$

## F  APPENDIX FOR SECTION 5

In this section, we present supplementary material to Section 5. In particular, we present the comparison of the result of `Confident Continuous Threshold Greedy` in Theorem 5 to the Accelerated Continuous Greedy algorithm (ACG) in Badanidiyuru & Vondrák (2014). Then in Section F, we provide detailed proof of Theorem 5. In addition, we provide the psedocode of `Confident Continuous Threshold Greedy` in Algorithm 5.

### F.1  Comparison of CCTG with Accelerated Continuous Greedy algorithm

In this section, we compare the results of Theorem 5 and the Accelerated Continuous Greedy algorithm (ACG) as presented in Badanidiyuru & Vondrák (2014).

1. First of all, we consider the case where we have exact access to the value oracle. In this case, we can get that $\widetilde{\Delta f}(S, s) = \Delta f(S, s) \leq \max_{s \in S} f(s)$ for any subset $S \subseteq U$ and element $s \in U$. This implies that $R$ can be set to be $\max_{s \in S} f(s)$. Consequently, from Theorem 5, the output solution set of CCTG satisfies that $f(S) \geq (1 - 1/e - O(\epsilon))f(OPT)$, which aligns with the approximation ratio presented in Badanidiyuru & Vondrák (2014). For the result on sample complexity, notice that each call of CS takes at most $\min\{O(\frac{\kappa}{\epsilon^2} \log \frac{n}{\delta\epsilon}), O(\frac{\kappa}{\epsilon\phi_X''} \log \frac{n}{\delta\epsilon\phi_X''})\}$ number of samples, where the first result is obtained by considering the worst case sample complexity of a fixed $\epsilon$-approximation. Since there are at most $\frac{3n}{\epsilon^2} \log \frac{\kappa}{\epsilon}$ calls of CS during CCTG, if we only consider the worst-case sample complexity, the total required sample complexity is at most $O(\frac{\kappa n}{\epsilon^3} \log^2 \frac{n}{\epsilon})$ for CCTG. This matches the result in Badanidiyuru & Vondrák (2014). In this sense, we improve the sample complexity when reduced to the case of assuming an exact oracle to the marginal gains.

2. On the other hand, from Theorem 5, we can see that even if the access to $\Delta f$ is noisy, as long as the upper bound on the noisy marginal gain $R$ is less than $f(OPT)$, the above analysis on sample complexity and approximation ratio holds. Hence, we can conclude that compared to access to an exact value oracle, the assumption of access to noisy marginal gain does not lead to additional sample complexity or a deterioration in the approximation ratio when compared to the scenario with an exact value oracle.

### F.2  Proof of Theorem 5

In this section, we present the detailed proof of Theorem 5 about our algorithm CCTG.

**Theorem 5.** *CCTG makes at most $\frac{3n}{\epsilon^2} \log \frac{3\kappa}{\epsilon}$ calls of CS. In addition, with probability at least $1 - \delta$, the following statements hold:*

- *The output fractional solution $\boldsymbol{x}$ achieves the approximation guarantee of $\boldsymbol{F}(\boldsymbol{x}) \geq (1 - e^{-1} - 2\epsilon)f(OPT) - R\epsilon$.*
- *Each call of CS on input $(w, \frac{\epsilon R}{2\kappa}, \frac{\delta\epsilon}{2nh'(\epsilon)}, \mathcal{D}_X, R)$ requires at most the minimum between*

$$\frac{18\kappa}{\epsilon^2} \log\left(\frac{8nh'(\epsilon)}{\delta\epsilon}\right)$$

*and*

$$\frac{36R}{\epsilon\phi_X''} \log\left(\frac{144R}{\epsilon\phi_X''} \sqrt{\frac{nh'(\epsilon)}{\delta\epsilon}}\right)$$

*noisy queries to the marginal gain. Here $OPT$ is an optimal solution to the MSMM problem, $\phi_X'' = \frac{\frac{\epsilon R}{2\kappa} - \epsilon\mathbb{E}X/3 + |w - \mathbb{E}X|}{2}$, and $h'(\epsilon) = \frac{3}{\epsilon} \log(\frac{3\kappa}{\epsilon})$.*

*Proof.* The second result on the sample complexity of calling the subroutine algorithm CS can be obtained immediately by applying the second result in (2a) in Lemma 17. Here we prove the first result in the theorem. Let us denote the fractional solution at time step $t$ as $\mathbf{x}_t$. From Lemma 18, it follows that conditioned on the events in Lemma 17, we have

$$\mathbf{F}(\mathbf{x}_{t+1}) - \mathbf{F}(\mathbf{x}_t) \geq \epsilon(1 - \epsilon)f(OPT)$$
$$- \epsilon(1 - \epsilon)\mathbf{F}(\mathbf{x}_{t+1}) - \epsilon^2 R.$$

It then follows that

$$\mathbf{F}(\mathbf{x}_{t+1}) \geq \frac{\mathbf{F}(\mathbf{x}_t) + \epsilon(1 - \epsilon)f(OPT) - \epsilon^2 R}{1 + \epsilon(1 - \epsilon)}$$
$$\geq (1 - \epsilon)\mathbf{F}(\mathbf{x}_t) + \epsilon(1 - \epsilon)^2 f(OPT) - \epsilon^2 R$$

---

**Algorithm 5:** `Confident Continuous Threshold Greedy` (CCTG)

---

1: **Input:** $\epsilon, \delta, \mathcal{M} \in 2^U$
2: $\mathbf{x} \leftarrow \mathbf{0}$
3: **for all** $s \in U$ and $s \in \mathcal{M}$ **do**
4:    $\hat{f}(s) \leftarrow$ sample mean over $\frac{18\kappa}{\epsilon^2} \log \frac{4n}{\delta}$ samples from $\mathcal{D}(\emptyset, s)$
5: **end for**
6: $d := \max_{s \in \mathcal{M}} \hat{f}(s)$,
7: **for** $t = 1$ to $1/\epsilon$ **do**
8:    $B \leftarrow$ Decreasing-Threshold Procedure $(\mathbf{x}, \epsilon, \delta, d, \mathcal{M})$
9:    $\mathbf{x} \leftarrow \mathbf{x} + \epsilon \cdot \mathbf{1}_B$
10: **end for**
11: **return x**

---

**Algorithm 6:** `Decreasing-Threshold Procedure` (DTP)

---

1: **Input:** $\mathbf{x}, \epsilon, \delta, d, \mathcal{M} \in 2^U$
2: $w \leftarrow d, B \leftarrow \emptyset$
3: **while** $w > \frac{\epsilon d}{3\kappa}$ **do**
4:    **for all** $u \in U$ **do**
5:      **if** $B \cup \{u\} \in \mathcal{M}$ **then**
6:        $X = \widetilde{\Delta} f(S(\mathbf{x} + \epsilon \mathbf{1}_B), u)$
7:        thre = `Confident Sample` $(w, \frac{R\epsilon}{2\kappa}, \frac{\delta\epsilon}{2nh'(\epsilon)}, \mathcal{D}_X, R)$
8:        **if** thre **then**
9:          $B \leftarrow B \cup \{u\}$
10:        **end if**
11:      **end if**
12:    **end for**
13:    $w = w(1 - \epsilon/3)$
14: **end while**
15: **return** $B$

---

Since there are $1/\epsilon$ iterations in CCTG, the output $\mathbf{x}$ satisfies that $\mathbf{x} = \mathbf{x}_{1/\epsilon}$. By applying induction to the above inequality, we would get

$$
\begin{aligned}
\mathbf{F}(\mathbf{x}_{1/\epsilon}) &\geq (1 - (1 - \epsilon)^{1/\epsilon})\{(1 - \epsilon)^2 f(OPT) - \epsilon R\} \\
&\geq (1 - 1/e)\{(1 - \epsilon)^2 f(OPT) - \epsilon R\} \\
&\geq (1 - 1/e - 2\epsilon) f(OPT) - \epsilon R.
\end{aligned}
$$

$\square$

**Lemma 17.** *With probability at least $1 - \delta$, the following two events hold.*

   *1. $(1 - \epsilon/3) \max_{s \in U} f(s) - \frac{R\epsilon}{2\kappa} \leq d \leq (1 + \epsilon/3) \max_{s \in U} f(s) + \frac{R\epsilon}{2\kappa}$.*

   *2. During each call of CS on the input $(w, \frac{\epsilon R}{2\kappa}, \frac{\delta\epsilon}{2nh'(\epsilon)}, \mathcal{D}_X, R, \epsilon/3)$ with the evaluated random variable being $X = \widetilde{\Delta} f(S(\boldsymbol{x} + \epsilon \boldsymbol{1}_B), u)$ where $\boldsymbol{x}$ is the fractional solution , $B$ is the set of coordinates and $u$ is an element in $U$, the results in Theorem 2 holds. I.e.,*

     *(a) CS takes at most the minimum between*

$$
\frac{18\kappa}{\epsilon^2} \log \left( \frac{8nh'(\epsilon)}{\delta\epsilon} \right)
$$

       *and*

$$
\frac{36R}{\epsilon\phi''_X} \log \left( \frac{144R}{\epsilon\phi''_X} \sqrt{\frac{nh'(\epsilon)}{\delta\epsilon}} \right).
$$

*(b) If the output is true, then*

$$(1 + \epsilon/3)\boldsymbol{E}\widetilde{\Delta f}(S(\boldsymbol{x} + \epsilon\boldsymbol{1}_B), u) \geq w - \frac{\epsilon R}{2\kappa}.$$

*If the output is false, then*

$$(1 - \epsilon/3)\boldsymbol{E}\widetilde{\Delta f}(S(\boldsymbol{x} + \epsilon\boldsymbol{1}_B), u) \leq w + \frac{\epsilon R}{2\kappa}.$$

*Proof.* First of all, by applying the inequality in Lemma 20, we have that for each fixed $s \in U$, after taking $N_4 = \frac{18\kappa}{\epsilon^2}\log\frac{4n}{\delta}$ number of samples, it follows that

$$P\big(|\hat{f}_{N_4}(s) - f(s)| \geq \frac{\epsilon}{3}f(s) + \frac{R\epsilon}{2\kappa}\big) \leq \frac{\delta}{2n}.$$

Taking a union bound over all elements in $U$, it follows that

$$P\big(|\hat{f}_{N_4}(s) - f(s)| \geq \frac{\epsilon}{3}f(s) + \frac{R\epsilon}{2\kappa}, \forall s \in U\big) \leq \frac{\delta}{2}.$$

Following the similar idea as in the proof of the Lemma 8, we can prove the first result.

Now we start to prove the second result. For each fixed call of CS with input $(w, \frac{\epsilon R}{2\kappa}, \frac{\delta\epsilon}{2nh'(\epsilon)}, \mathcal{D}_X,$ $R, \epsilon/3$ ), by applying the results in Theorem 2, we have that with probability at least $1 - \frac{\delta\epsilon}{2nh'(\epsilon)}$, both the statements about the sample complexity in (2a) and approximation guarantee in (2b) in the lemma holds. Since there are $1/\epsilon$ calls of the Decreasing-Threshold Procedure and each Decreasing-Threshold Procedure makes at most $nh'(\epsilon)$ calls of the CS algorithm, there are at most $nh'(\epsilon)/\epsilon$ calls of the CS algorithm. By taking the union bound, we can prove that with probability at least $1 - \delta/2$, the second results hold. By taking the union bound again, we can see that with probability at least $1 - \delta$, all of the results in the lemma hold. $\square$

**Lemma 18.** *Conditioned on the two events defined in Lemma 17, we have that during each implementation of* Decreasing-Threshold Procedure, *the output coordinate set $B$ satisfies that*

$$\boldsymbol{F}(\boldsymbol{x} + \epsilon\boldsymbol{1}_B) - \boldsymbol{F}(\boldsymbol{x}) \geq \epsilon(1 - \epsilon)\{f(OPT) - \boldsymbol{F}(\boldsymbol{x} + \epsilon\boldsymbol{1}_B)\}$$
$$- \epsilon^2 R.$$

*Proof.* Here we denote the output solution set as $B = \{b_1, b_2, ..., b_\kappa\}$ where $b_i$ is the $i$-th element that is added to set $B$. Here if $|B| < \kappa$, then for any $i > |B|$, $b_i$ is defined as a dummy variable. Since $\mathcal{M}$ is a matroid, there exists a permutation of the optimal solution $OPT = \{o_1, o_2, ..., o_\kappa\}$ such that $B_{i-1} \cup \{o_i\} \in \mathcal{M}$ for each $i \in [\kappa]$. For notation simplicity, we also define $G(\mathbf{x}, u) = \boldsymbol{E}\widetilde{\Delta f}(S(\mathbf{x}), u)$. First of all, we prove the following claim: for each $i \in [\kappa]$, we have that

$$G(\mathbf{x} + \epsilon\boldsymbol{1}_{B_{i-1}}, b_i) \geq (1 - \epsilon)G(\mathbf{x} + \epsilon\boldsymbol{1}_{B_{i-1}}, o_i) - \frac{\epsilon R}{\kappa}$$

The proof is as follows: if the element $b_i$ is added at the first iteration, then from Lemma 17, we have that $(1 + \epsilon/3)G(\mathbf{x} + \epsilon\boldsymbol{1}_{B_{i-1}}, b_i) \geq w - \frac{\epsilon R}{2\kappa}$. Since the threshold at the first iteration is $w = d$, and $d \geq (1 - \epsilon/3)\max_{s \in U} f(s) - \frac{R\epsilon}{2\kappa}$ according to the first result in Lemma 17, then

$$(1 + \epsilon/3)G(\mathbf{x} + \epsilon\boldsymbol{1}_{B_{i-1}}, b_i) \geq (1 - \epsilon/3)\max_{s \in U} f(s) - \frac{\epsilon R}{\kappa}.$$

Since $\max_{s \in U} f(s) \geq \max_{o \in OPT} f(o) \geq G(\mathbf{x} + \epsilon\boldsymbol{1}_{B_{i-1}}, o_i), \forall i \in [\kappa]$, it then follows that

$$G(\mathbf{x} + \epsilon\boldsymbol{1}_{B_{i-1}}, b_i) \geq (1 - \epsilon)G(\mathbf{x} + \epsilon\boldsymbol{1}_{B_{i-1}}, o_i) - \frac{\epsilon R}{\kappa}.$$

If $b_i$ is not a dummy variable and is not added in the first iteration, we can see that $(1 + \epsilon/3)G(\mathbf{x} + \epsilon\boldsymbol{1}_{B_{i-1}}, b_i) \geq w - \frac{R\epsilon}{2\kappa}$. Since the element $o_i$ is not added to $B$, it is not added at the last iteration. By the construction of $OPT$, we have that $B_{i-1} \cup \{o_i\} \in \mathcal{M}$. Therefore,

$$(1 - \epsilon/3)G(\mathbf{x} + \epsilon\boldsymbol{1}_{B_{i-1}}, o_i) \leq \frac{w}{1 - \epsilon/3} + \frac{R\epsilon}{2\kappa}.$$

Then

$$G(\mathbf{x} + \epsilon \mathbf{1}_{B_{i-1}}, b_i) \geq \frac{(1 - \epsilon/3)^2 G(\mathbf{x} + \epsilon \mathbf{1}_{B_{i-1}}, o_i)}{1 + \epsilon/3}$$

$$- \frac{(1 - \epsilon/3)\epsilon R}{2(1 + \epsilon/3)\kappa} - \frac{R\epsilon}{2(1 + \epsilon/3)\kappa}$$

$$\geq (1 - \epsilon)G(\mathbf{x} + \epsilon \mathbf{1}_{B_{i-1}}, o_i) - \frac{\epsilon R}{\kappa}.$$

Next, we consider the case where $b_i$ is a dummy variable. In this case $G(\mathbf{x} + \epsilon \mathbf{1}_{B_{i-1}}, b_i) = 0$. Since $o_i$ is not added,

$$(1 - \epsilon/3)G(\mathbf{x} + \epsilon \mathbf{1}_{B_{i-1}}, o_i) \leq \frac{\epsilon d}{3\kappa} + \frac{R\epsilon}{2\kappa}.$$

Since $d \leq (1 + \epsilon/3) \max_{s \in U} f(s) + \frac{R\epsilon}{2\kappa} \leq (1 + \epsilon/3)R + \frac{R\epsilon}{2\kappa}$. Notice that when $\epsilon > 0.5$, the approximation guarantee in Theorem 5 is trivial. Therefore, here we can assume $\epsilon \leq 0.5$, which implies that $d \leq 3R/2$. Then we have that

$$(1 - \epsilon/3)G(\mathbf{x} + \epsilon \mathbf{1}_{B_{i-1}}, o_i) \leq \epsilon R/\kappa.$$

Therefore,

$$G(\mathbf{x} + \epsilon \mathbf{1}_{B_{i-1}}, b_i) = 0$$

$$\geq (1 - \epsilon/3)G(\mathbf{x} + \epsilon \mathbf{1}_{B_{i-1}}, o_i) - \epsilon R/\kappa.$$

With this claim, we can prove the results of the lemma.

$$\mathbf{F}(\mathbf{x} + \epsilon \mathbf{1}_B) - \mathbf{F}(\mathbf{x}) = \sum_{i=1}^{\kappa} \mathbf{F}(\mathbf{x} + \epsilon \mathbf{1}_{B_i}) - \mathbf{F}(\mathbf{x} + \epsilon \mathbf{1}_{B_{i-1}})$$

$$= \sum_{i=1}^{\kappa} \epsilon \cdot \frac{\partial \mathbf{F}}{\partial b_i} \big|_{x = \mathbf{x} + \mathbf{1}_{B_{i-1}}}$$

$$\geq \epsilon \sum_{i=1}^{\kappa} \boldsymbol{E} \Delta f(S(\mathbf{x} + \epsilon \mathbf{1}_{B_{i-1}}), b_i)$$

$$= \epsilon \sum_{i=1}^{\kappa} G(\mathbf{x} + \epsilon \mathbf{1}_{B_{i-1}}, b_i).$$

Here the last equality comes from the fact that $\mathbb{E}\Delta f(S(\mathbf{x}), u) = \mathbb{E}\widetilde{\Delta f}(S(\mathbf{x}), u)$. By the claim, it follows that

$$\mathbf{F}(\mathbf{x} + \epsilon \mathbf{1}_B) - \mathbf{F}(\mathbf{x}) \geq \epsilon \sum_{i=1}^{\kappa} (1 - \epsilon)G(\mathbf{x} + \epsilon \mathbf{1}_{B_{i-1}}, o_i) - \epsilon^2 R$$

$$= \epsilon(1 - \epsilon) \sum_{i=1}^{\kappa} \boldsymbol{E} \Delta f(S(\mathbf{x} + \epsilon \mathbf{1}_{B_{i-1}}), o_i)$$

$$- \epsilon^2 R$$

$$\geq \epsilon(1 - \epsilon) \sum_{i=1}^{\kappa} \boldsymbol{E} \Delta f(S(\mathbf{x} + \epsilon \mathbf{1}_B), o_i)$$

$$- \epsilon^2 R$$

$$\geq \epsilon(1 - \epsilon)\{f(OPT) - \mathbf{F}(\mathbf{x} + \epsilon \mathbf{1}_B)\}$$

$$- \epsilon^2 R.$$

Here the second and third inequality are due to submodularity and monotonicity. □

## G  Technical Lemmas

**Lemma 19** (Hoeffding's Inequality). *Let $X_1, ..., X_N$ be independent random variables such that $X_i$ is $R$-sub-Gaussian and $\mathbb{E}[X_i] = \mu$ for all $i$. Let $\overline{X} = \frac{1}{N} \sum_{i=1}^{N} X_i$. Then for any $t > 0$,*

$$P(|\overline{X} - \mu| \geq t) \leq 2 \exp\{-\frac{Nt^2}{2R^2}\}.$$

**Lemma 20** (Relative + Additive Chernoff Bound (Lemma 2.3 in Badanidiyuru & Vondrák (2014))). *Let $X_1, ..., X_N$ be independent random variables such that for each $i$, $X_i \in [0, R]$ and $\mathbb{E}[X_i] = \mu$ for all $i$. Let $\widehat{X}_N = \frac{1}{N} \sum_{i=1}^{N} X_i$. Then*

$$P(|\widehat{X}_N - \mu| > \alpha\mu + \epsilon) \leq 2 \exp\{-\frac{N\alpha\epsilon}{3R}\}.$$

**Lemma 21.** *Let $X_1, ..., X_N$ be independent random variables such that $X_i \in [0, R]$ and $\mathbb{E}[X_i] = \mu$ for all $i$. Let $\overline{X} = \frac{1}{N} \sum_{i=1}^{N} X_i$. Then for any $t > 0$ and $\delta > 0$, if*

$$N \geq \frac{R^2 \ln(1/\delta)}{t^2},$$

*then $P(|\overline{X} - \mu| \geq t) \leq \delta$.*

*Proof.* This result follows easily from Hoeffding's Inequality. □

**Lemma 22.** *Let $X_1, ..., X_N$ be independent random variables such that $X_i \in [0, R]$ and $\mathbb{E}[X_i] = \mu$ for all $i$. Let $\overline{X} = \frac{1}{N} \sum_{i=1}^{N} X_i$. Then for any $\delta > 0$, if*

$$c \geq R\sqrt{\frac{\ln(2/\delta)}{2N}}, \tag{6}$$

*it is the case that*

$$P(\mu \in [\overline{X} - c, \overline{X} + c]) \leq \delta.$$

*Proof.* This result follows easily from Hoeffding's Inequality. □

**Lemma 23.** *Suppose $x \in \mathbb{R}$ and $x \geq 2$, if we have $x \geq \frac{2}{a} \log \frac{2}{a}$, then it holds that*

$$\frac{\log x}{x} \leq a$$

*Proof.* Since $y = \frac{\log x}{x}$ is decreasing when $x \geq 2$, if $x > \frac{2}{a} \log \frac{2}{a}$, then we have

$$\frac{\log x}{x} < \frac{a}{2} \cdot \frac{\log(\frac{2}{a} \log \frac{2}{a})}{\log \frac{2}{a}} \leq a.$$

□

## H  Additional Experiments

In this section, we present some additional details of our experiments. In particular, we present additional detail about the experimental setup in Section H.1. Next, we present the additional experimental results in Section H.2.

### H.1  Additional experimental setup

First of all, we provide details about the two applications used to evaluate our algorithms. The two applications considered here are noisy data summarization as presented in Section H.1.1 and influence maximization in Section H.1.2.

### H.1.1 Noisy data summarization

In data summarization, $U$ is a dataset that we wish to summarize by choosing a subset of $U$ of cardinality at most $\kappa$. The objective function $f : 2^U \rightarrow \mathbb{R}_{\geq 0}$ takes a subset $X \subseteq U$ to a measure of how well $X$ summarizes the entire dataset $U$, and in many cases is monotone and submodular Tschiatschek et al. (2014). However, in real instances of data summarization, we may not have access to an exact measure $f$ of the quality of a summary, but instead, we may have authentic human feedback which is modeled as noisy queries to some underlying monotone and submodular function Singla et al. (2016).

Motivated by this, we run our experiments using instances of noisy data summarization. Our underlying monotone submodular function $f$ is defined as follows: $U$ is assumed to be a labeled dataset, e.g. images tagged with descriptive words, and for any $X \subseteq U$, $f$ takes $X$ to the total number of tags represented by at least one element in $X$ Crawford (2023). Notice that this is essentially the instance of set cover.

### H.1.2 Influence maximization

Another application is the influence maximization problem in large-scale networks Kempe et al. (2003). In this application, the universe is the set of users in the social network, and the objective is to choose a subset of users to seed with a product to advertise in order to maximize the spread throughout the network. The marginal gain of adding an element $s$ to set $S$ is defined as $\Delta f(S, s) := \mathbb{E}_{\mathbf{w} \sim \mathcal{D}(\bar{\mathbf{w}})} \Delta f(S, s; \mathbf{w})$, where $\mathbf{w}$ is the noisy realization of the graph from some unknown distribution $\mathcal{D}(\bar{\mathbf{w}})$, and $\Delta f(S, s; \mathbf{w}) = f(S \cup \{s\}; \mathbf{w}) - f(S; \mathbf{w})$. In a noisy graph realization with parameter $\mathbf{w}$, $f(S; \mathbf{w})$ is the number of elements influenced by the set $S$ under some influence cascade model. It is #P-hard to evaluate the objective in influence maximization Chen et al. (2010). Many of the previous works Chen et al. (2009) assume the entire graph can be stored by the algorithm and the influence cascade model is known. The algorithm first samples some graph realizations to approximate the true objective and run submodular maximization algorithms on the sampled graphs. In contrast, our setting and algorithm do not assume that a graph is stored or the model of influence is explicitly known, only that we could simulate it for a subset. Therefore our approach could apply in more general influence maximization settings than the sampled realization approach.

Next, we describe the details about the three algorithms that we compare to: (i) The fixed $\epsilon$ approximation ("EPS-AP") algorithm. This is where we essentially run CTG, except instead of using the subroutine CS to adaptively sample in order to reduce the number of samples, we simply sample down to an $\epsilon$-approximation of every marginal gain. This takes $N_1$ samples for every marginal gain computation, see definition of $N_1$ in Algorithm 1. The element $u$ is added to $S$ if and only if the empirical estimate $\widehat{\Delta f_{N_1}}(S, u) \geq w$; (ii) The special case of the algorithm ExpGreedy of Singla et al. (2016) that yields about a $(1 - 1/e)$-approximate solution with high probability, "EXP-GREEDY", which is described in Section 1.1 and in the appendix. In the detailed description of ExpGreedy found in the appendix in the supplementary material, this is the case that $k'$ is set to be 1; (iii) The randomized version of the algorithm of ExpGreedy, "EXP-GREEDY-K", which yields about a $(1 - 1/e)$-approximation guarantee in expectation. Since EXP-GREEDY-K is a randomized algorithm, we average the results for EXP-GREEDY-K over 10 trials. This is the case that $k' = \kappa$.

Then we provide some additional details for experiments on instances of data summarization. The parameter $\delta$ for all the experiments is set to be 0.2, and the approximation precision parameter $\alpha$ is 0.2 for both CTG and EPS-AP. The value of $\epsilon$ of the experiments for different $\kappa$ are 0.1, 0.2, 0.1 and 0.1 on corel_60, delicious_300, delicious, and corel respectively. The value of $\kappa$ for different $\epsilon$ are 10, 80, 200 and 100 on corel_60, delicious_300, delicious and corel respectively.

At last, we introduce the experimental setup for influence maximization. We run the four algorithms described above on the experiments for different values of $\kappa$ and $\epsilon$. The dataset used here is a subgraph extracted from the EuAll dataset with $n = 29$ Leskovec & Sosič (2016). The underlying weight of each edge is uniformly sampled from $[0, 1]$ ("euall"). In our experiments, we simulate the influence maximization under the influence cascade model. We further use the reverse influence sampling (RIS) Borgs et al. (2014) to enhance the computation efficiency of our algorithm. Here $R$ is the number of nodes in the graph and is thus 29. The value of $\kappa$ for different $\epsilon$ is 8, and the value of $\epsilon$ for different $\kappa$ is 0.15. The parameters $\delta$ and $\alpha$ are set to be 0.2 for both of the experiments. Since EXP-GREEDY-K

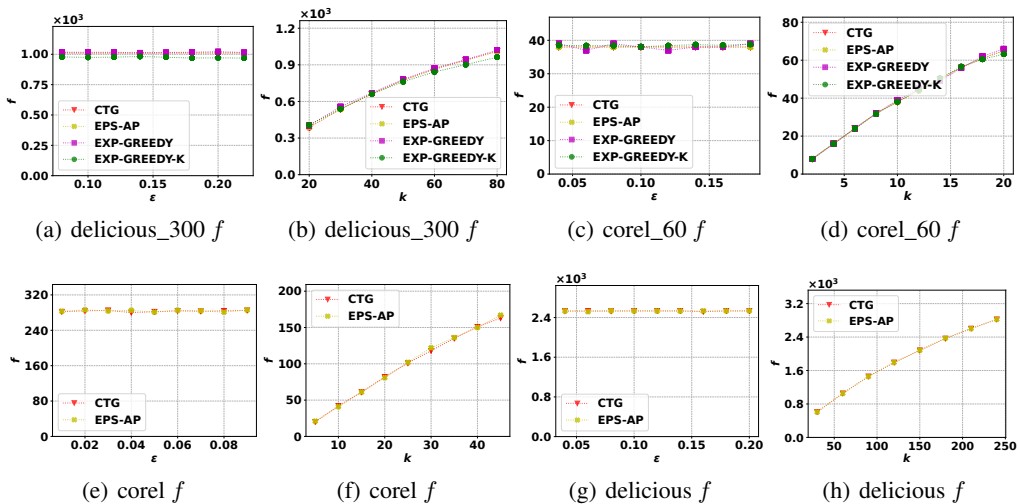

Figure 4: The experimental results of $f$ of running different algorithms on instances of data summarization on the delicious URL dataset ("delicious", "delicious_300") and Corel5k dataset ("corel", "corel_60").

is a randomized algorithm, the experimental results for `EXP-GREEDY-K` are averaged over 4 trials for different $\epsilon$, and 8 trials for different $\kappa$.

## H.2 ADDTIONAL EXPERIMENTAL RESULTS

First, we present the result analysis of the experiments where we vary $\epsilon$. It can be seen from Figures 3(a), 3(b), 3(e) and 3(f) that both the total samples and average samples of our algorithm `CTG` increase less compared with `EPS-AP` and `EXP-GREEDY` as $\epsilon$ decreases. This is not surprising, because the theoretical guarantee on the number of samples taken per marginal gain contribution in `EPS-AP` is $O(\frac{1}{\epsilon^2})$, which would increase rapidly when $\epsilon$ decreases. This also makes sense for `EXP-GREEDY`, since the theoretical guarantee on the number of queries of each iteration is $O(\frac{nR^2}{\epsilon^2} \log \left( \frac{R^2 kn}{\delta \epsilon^2} \right))$ if the difference between elements marginal gains are very small.

Then we present the additional experimental results with respect to the function value $f$ on the instance of data summarization in the main paper. The results are in Figure 4. The experimental results of $f$ for different $\kappa$ are in Figure 4(b), 4(h), 4(d) and 4(f). From the results, one can see that the $f$ values for different algorithms are very almost the same in most cases. However, when $\kappa$ increases and becomes large, the $f$ value of `EXP-GREEDY-K` is smaller than other algorithms, which is because when $\kappa$ is large, it allows for more randomness in `EXP-GREEDY-K` and is less accurate.

Next, we present the experimental results on the instance of influence maximization. The results are plotted in Figure 5. From the results, we can see that our proposed algorithm `CTG` outperforms the other three algorithms in terms of the total number of samples (see Figure 5(a), 5(d)). When $\kappa$ increases, the average number of samples decreases fast for `CTG`. This is because the marginal gain on this instance decreases rapidly when $\kappa$ increases while the threshold value decreases only by a factor of $1 - \alpha$ at the end of each iteration, in many iterations the threshold value $w$ is much higher than the marginal gain and thus the gap function $\phi(S, s)$ is large. According to the results of sample complexity in Theorem 3, the number of required samples decreases fast as $\kappa$ increases. This is also why the average number of samples of `CTG` is much smaller than `EXP-GREEDY` and `EXP-GREEDY-K` as is presented in Figure 5(b) and Figure 5(e).

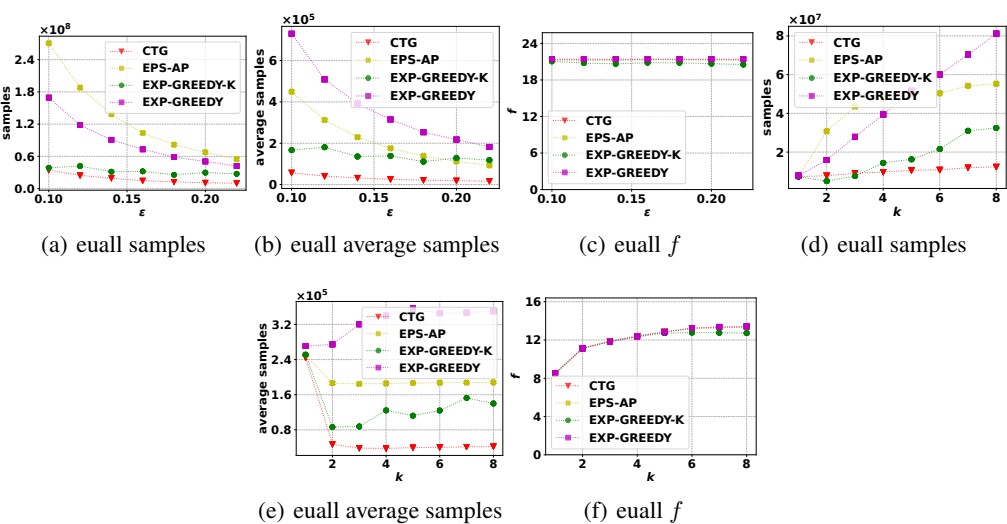

Figure 5: The experimental results of running different algorithms on the instance of influence maximization on the EuAll dataset ("euall").

