# OpenReview forum: "Adaptive Threshold Sampling for Fast Noisy Submodular Maximization"
_ICLR.cc/2025/Conference — Submitted to ICLR 2025_

### Official Review · Reviewer_XgDq · 2024-10-20

**Soundness:** 2
**Presentation:** 2
**Contribution:** 2
**Rating:** 6
**Confidence:** 3

**Summary:**

The paper introduces the Confident Sample (CS) algorithm to maximize submodular functions under noisy conditions efficiently. By leveraging insights from multi-armed bandit algorithms, CS reduces the number of noisy queries required to approximate submodular functions, making it applicable to diverse optimization tasks such as influence maximization and recommendation systems. Theoretical analysis and empirical results demonstrate the effectiveness of CS in achieving competitive approximation guarantees with significantly improved sample efficiency compared to traditional methods.

**Strengths:**

The paper proposes the Confident Sample (CS) algorithm, which effectively reduces the number of noisy queries by dynamically adjusting the sample size according to the level of uncertainty. This adaptive approach contrasts with traditional fixed-precision methods, offering substantial improvements in sample efficiency. The work's theoretical contributions are robust, providing guarantees on both approximation quality and sample complexity, making it a competitive alternative to existing methods like ExpGreedy. These theoretical insights are further supported by empirical evaluations on real-world datasets, where the proposed algorithms demonstrate superior sample efficiency, highlighting the practical relevance of the approach.

**Weaknesses:**

I do not have significant negative comments regarding the contributions of this paper. My only concern is whether the topic aligns well with ICLR's scope, as I have not come across purely theoretical work on submodular maximization algorithms published at ICLR before. While submodular maximization is indeed a crucial problem in machine learning, ICLR, to my knowledge, tends to focus more on areas related to deep learning and neural networks. Therefore, would it be more appropriate to consider submitting this paper to venues like NeurIPS or ICML, which might better align with its focus?

**Questions:**

Please refer to weaknesses.

---

> ### Author Response · Authors · 2024-11-23
> **Author Rebuttal**
>
> > I do not have significant negative comments regarding the contributions of this paper. My only concern is whether the topic aligns well with ICLR's scope, as I have not come across purely theoretical work on submodular maximization algorithms published at ICLR before. While submodular maximization is indeed a crucial problem in machine learning, ICLR, to my knowledge, tends to focus more on areas related to deep learning and neural networks. Therefore, would it be more appropriate to consider submitting this paper to venues like NeurIPS or ICML, which might better align with its focus?
>
> Submodular optimization problems are important for many applications in machine learning, such as exemplar-based clustering, influence maximization, and reinforcement learning. While it is the case that more papers on the topic of submodularity appear at NeurIPS and ICML, this topic has recently been of interest at ICLR. In particular, there are a total of 7 submissions to ICLR 2025 that are on submodular optimization (many of which are mainly focused on algorithms), and 3 papers appeared at ICLR 2024 on the topic. As one example, the paper [1] is no less theoretical than our own work. Besides, another topic related to our paper is the multi-armed bandit problem under a pure-exploration setting. There are also many papers on this topic that have been published in ICLR. For example, [2] and [3] are published in ICLR 2024 and the main focus of the mentioned papers is theoretical.
>
> [1] Pedramfar, Mohammad, et al. "Unified Projection-Free Algorithms for Adversarial DR-Submodular Optimization." ICLR, 2024.
> [2] Zixin Zhong, Wang Chi Cheung, Vincent Y. F. Tan. "Achieving the Pareto Frontier of Regret Minimization and Best Arm Identification in Multi-Armed Bandits", ICLR, 2024.
> [3] Zhirui Chen, P. N. Karthik, Yeow Meng Chee, and Vincent Y. F. Tan. "Fixed-Budget Differentially Private Best Arm Identification", ICLR, 2024.

---

> > ### Comment · Reviewer_XgDq · 2024-11-23
> >
> > Thank you for your response. After considering the comments from other reviewers, I have decided to maintain my original rating.

---

### Official Review · Reviewer_KUeE · 2024-10-29

**Soundness:** 3
**Presentation:** 2
**Contribution:** 2
**Rating:** 5
**Confidence:** 4

**Summary:**

This paper studies submodular maximization in a noisy model. Formally, the goal is the same as in the standard monotone submodular maximization with cardinality or matroid constraints, but the algorithm has access to the function via a noisy oracle. In particular, given a set $S$ and element $x$, the algorithm can query $(S,x)$ to the oracle and receives a random (unbiased) estimator of the marginal value of $x$ with respect to $S$. The authors assume that such estimator is subgaussian (of parameter $R$ that is known).

The paper's contribution is to provide tight approximation results for cardinality and matroid constraints by adapting known techniques with a sample efficient estimation procedure called Confident Sample (CS).

**Strengths:**

Submolar maximization is a relevant topic for the NeurIPS/ICML/ICLR audience, given its vast applicability in ML. The model of noisy queries is natural and well-motivated. The sample complexity bounds are not trivial (as the authors explain, using Hoeffding would already yield some results).

**Weaknesses:**

- The technical contribution is moderate. In the end, the paper's contribution lies in rewriting concentration bounds, parameterized by a notion of gap. Threshold-based algorithms are well-known in the submodular literature.
- The sample complexity bounds are pretty involved. Many parameters are entailed, and a clear picture is difficult to get.

**Questions:**

Do you have any tightness results on the sample complexity?

---

> ### Author Response · Authors · 2024-11-23
> **Author Rebuttal**
>
> > The technical contribution is moderate. In the end, the paper's contribution lies in rewriting concentration bounds, parameterized by a notion of gap. Threshold-based algorithms are well-known in the submodular literature.
>
> We do not simply rewrite concentration bounds parameterized by a notion of gap. Our sampling strategy is fundamentally different from applying a concentration bound to a single batch of i.i.d. samples and is significantly more sample-efficient. In standard applications of concentration inequalities such as Hoeffding's inequality in submodular optimization, a fixed batch of samples is used.  It isn't possible to sample until we've reached the gap because there is only a single batch of samples and we don't have any clue what $\phi$ would be before we take them. $\phi$ arises in the adaptive sampling context, where samples are taken one-by-one. This may seem similar to the former approach, but there are significant technical challenges that arise as a result. We describe this more thoroughly in the response to all reviewers, and we have included this discussion in the updated version of our paper highlighted in blue in the appendix.
>
>
> > The sample complexity bounds are pretty involved. Many parameters are entailed, and a clear picture is difficult to get.
>
> Let us first examine the sample complexity for a specific instance of CS. In Theorem 1, the term on the left-hand side, $\frac{2R^2}{\epsilon^2}\log \left(\frac{4}{\delta}\right)$, represents the number of samples required to approximate $X$ within $\epsilon$-distance with probability, i.e., $|X-\mathbb{E}X|\leq\epsilon$. This corresponds to case (d) in Figure 1, and is the number of samples that the fixed $\epsilon$-approximation would take. Such a large number of samples is only necessary when $\mathbb{E} X$ is close to the threshold, and therefore many samples are needed to see if it is above or below the threshold. Importantly, this value can be obtained without adaptive sampling.
>
> The value on the right-hand side of the sample compleixty result in Theorem 1 comes from the adaptive sampling, and it is the number of samples required to shrink the confidence interval just enough so that we can conclude whether $\mathbb{E}X$ is approximately above or below the threshold, and it depends on how far $\mathbb{E}X$ is from the threshold, i.e. the value of $\phi$ (since a larger gap allows for a wider confidence interval upon stopping and thus fewer samples. ). This latter value cannot be computed before we start sampling, and is a result of the adaptive sampling where we do not know how many samples we will take initially. This corresponds to cases (a) and (b) in Figure 1.
>
> Now consider Theorems 3, 4, and 5, which provide the sample complexity of CS within the algorithms. These results are derived by setting the failure probability $\delta$ in Theorems 1 or 2 to be the reciprocal of the total number of calls to CS, (i.e. the number of marginal gain queries) multiplied by $\delta'$. This adjustment ensures that, via a union bound, the overall algorithm succeeds with a probability of at least  $1-\delta$. We have incorporated this discussion into the updated version of the paper, highlighted in blue.

---

### Official Review · Reviewer_JMtB · 2024-11-07

**Soundness:** 3
**Presentation:** 3
**Contribution:** 2
**Rating:** 5
**Confidence:** 5

**Summary:**

The paper studies the constrained submodular maximization under noise problem that arises in many applications in AI and ML Communities. The key algorithm of this work is Confident Sample (CS), which is inspired by algorithms for best-arm-identification in multi-armed bandit. The CS algorithm can then be integrated into many existing approximation algorithms for submodular maximization under constraints. The authors show that the integrated algorithms take fewer samples on both theoretical and practical sides.

**Strengths:**

- This work studies an interesting and meaningful research problem in the AI ​​and ML community. The paper is well-written and structured.
- The core algorithm, Confident Sample (CS), has been shown to be effective in estimating the expectation of the objective submodular function in Gause distributions. The theoretical analysis is natural and reliable. However, the techniques for proving them are pretty elementary.

**Weaknesses:**

- Except for CS, the remaining algorithms are not new. The author's main contribution is how to apply CS algorithm to existing algorithms and the corresponding theoretical analysis.
- The idea of the CS algorithm is not new. It existed in previous algorithms (For example, in Alg 2 in [Mat]).
- It is natural to apply CS algorithms to existing algorithms, but it is not difficult to derive theoretical bounds.

======================

Ref.

[Mat] Matthew Fahrbach, Vahab S. Mirrokni, Morteza Zadimoghaddam: Submodular Maximization with Nearly Optimal Approximation, Adaptivity and Query Complexity. SODA 2019: 255-273.

**Questions:**

1. What are the differences (ideas, theoretical analysis) between CS and Alg 2 in [Mat]?
2. What are the challenges in getting better theoretical bounds when applying CS algorithms to existing algorithms?
3. Does the CS algorithm work well with other distributions?
4. Can the CS algorithm be applied to the Minimum Cost Submodular Cover (MCSC) problem? The paper's contribution would be better if it is possible to apply CS to MCSC with better theoretical bounds.

---

> ### Author Response · Authors · 2024-11-23
> **Author Rebuttal**
>
> > Except for CS, the remaining algorithms are not new. The author's main contribution is how to apply CS algorithm to existing algorithms and the corresponding theoretical analysis ... What are the challenges in getting better theoretical bounds when applying CS algorithms to existing algorithms?"
>
> > The idea of the CS algorithm is not new. It existed in previous algorithms (For example, in Alg 2 in [Mat]) ... What are the differences (ideas, theoretical analysis) between CS and Alg 2 in [Mat]?
>
> Algorithm 2 in [Mat] does not appear to be the same as the CS algorithm but is similar to what we call a fixed $\epsilon$-approximation in the paper (see Section 2 in our manuscript for a definition of this). In fact, the CS algorithm improves upon this approach.
>
> In particular, in Algorithm 2 [Mat] takes a single batch of samples in order to approximate the mean of their distribution $\mathcal{D}_t$.  In contrast, the key idea of CS is to sample one at a time and check the mean and confidence interval after every sample (i.e. sample "adaptively"), which introduces significant technical challenges in the development and analysis of CS. We give a more thorough description of why the analysis of our adaptive sampling approach is much different than algorithms such as that of [Mat] in the response to all reviewers. Here, we briefly summarize the key points. The technical challenges and differences include:
> - The fixed-$\epsilon$ approximation only use the concentration inequality for a single time on a fixed number of samples while CS algorithm requires applying the concentration inequality for each sample.
> - The fixed-$\epsilon$ approximation takes a predetermined number of samples while in the CS algorithm, the number of samples is a random variable dependent on the outcomes of previous samples. This caused difficulty in applying the concentration inequality.
> - In CS, the confidence interval is designed to adapt with each new sample, gradually shrinking as the number of samples increases (see Theorem 1). In addition, the failure probability is adjusted dynamically based on how many samples we've taken so far (see proof of Lemma 6).
> - In Theorem 2 and 4, we use a combination of Hoeffding and Chernoff that is well-suited to the threshold algorithms, which results in improved dependence on $R$ when $R$ is large .
>
>
>  It may in fact be possible that the algorithm of [Mat] could use something like CS as a subroutine, improving their sample complexity, which illustrates CS as a widely useful algorithm.
>
> [Mat] Matthew Fahrbach, Vahab S. Mirrokni, Morteza Zadimoghaddam: Submodular Maximization with Nearly Optimal Approximation, Adaptivity and Query Complexity. SODA 2019: 255-273.
>
> > "Confident Sample (CS), has been shown to be effective in estimating the expectation of the objective submodular function in Gause distributions ... Does the CS algorithm work well with other distributions?"
>
> The CS algorithm is for any random variable that is $R$-sub-Gaussian, which include bounded and Gaussian random variables.
>
> > "Can the CS algorithm be applied to the Minimum Cost Submodular Cover (MCSC) problem? The paper's contribution would be better if it is possible to apply CS to MCSC with better theoretical bounds."
>
> The CS algorithm works for a wide range of submodular algorithms which has the thresholding step to check whether the marginal gain is above the threshold or not. Therefore, if we adopt an algorithm based on threshold greedy algorithm, it is also possible for our algorithm to work for MCSC algorithms. For example, we can use bicriteria algorithms developed in [1].
>
> [1] Chen, Wenjing, and Victoria Crawford. "Bicriteria approximation algorithms for the submodular cover problem", NeurIPS, 2024.

---

> ### Comment · Reviewer_JMtB · 2024-11-23
> **Comment**
>
> Thanks for the feedback from the authors. I still think the paper is technically average. The CS algorithm is not really new, it borrows ideas from previous algorithms. Besides, I think that putting CS into existing algorithms without any improvements is not a significant contribution. In submodular function optimization, developing new techniques that improve theoretical bounds is more important. Therefore I keep my score of 5.

---

### Official Review · Reviewer_NxXo · 2024-11-09

**Soundness:** 3
**Presentation:** 3
**Contribution:** 2
**Rating:** 6
**Confidence:** 2

**Summary:**

The topic of this paper is the problem of submodular maximization:
given a ground set $U$ and a an oracle access to a
submodular objective function $f: U \to \mathbb{R}$,
our task is to find $S\subseteq U$ with the highest value of $f(S)$.
They study also the classical constrained versions where
we maximize $f(S)$ subject to cardinality and matroid constraints.

The authors study submodular maximization in the setting,
where
exact evaluation of $f(S)$ is not possible
and one can obtain only noisy estimate. In particular, they assume
the oracle returns a noisy estimate which is unbiased and
is $R$-subgaussian.
This setting was already studied before, e.g. by Singla et al. '15, who
achieved almost the same approximation guarantees as in the classical
non-noisy setting and provided bounds on the number of the performed
noisy queries.
Authors provide theoretical bounds which they compare to previous
works and they also present an empirical comparison.

The main difference between their work and Singla et al. '15
is that their algorithm is based on a faster implementation
of the classical greedy algorithm by Badanidiyuru and Vondrak which, instead
of comparing the marginal gain of the elements to each other, it compares
their marginal gain to a threshold chosen during algorithm's runtime.
Therefore, instead of a quadratic dependence on the gap $\Delta_{max}$
between the two top marginal gains, they have a quadratic dependence on the
gap from the threshold (their parameter $\phi$).

**Strengths:**

The problem is important and their setting seems relevant and well motivated.
They achieve improvements in sample complexity over previous works at least in
some settings. They have a better running time as well.

**Weaknesses:**

The results are somewhat difficult to appreciate for me,
despite that they provide more than a page long comparison with the
previous works in Appendix discussing the differences in the long formulas.
For example, it is not clear to me why is it better to have dependence
on $\phi$ instead of $\Delta_{max}$.
Is $\phi$ always smaller than $\Delta_{max}$?

I am not an expert in the field. While I see that the authors do achieve
an improvement over the previous works, I do not see its significance.
Therefore my rating.

**Questions:**

* Can you comment on the weakness above?

* Can you say that your algorithm has a better sample complexity
than the previous works always?

* Knowing something about the properties of the input instance, how can
you estimate the value of $\phi$ without running your algorithm?
I am asking this to understand whether your bounds can be used to predict your algorithm's performance. Note that, for example, $\Delta_{max}$ can be
estimated in advance based on the properties of the input instance.

---

> ### Author Response · Authors · 2024-11-22
> **Author Rebuttal**
>
> > The results are somewhat difficult to appreciate for me, despite that they provide more than a page-long comparison with [Singla et al. (2016)] in Appendix discussing the differences in the long formulas ... While I see that the authors do achieve an improvement over the previous works, I do not see its significance.
>
> One important difference is that the work of Singla et al. only considers the problem of monotone submodular maximization with a cardinality constraint. In contrast, our proposed CS algorithms could serve as a general tool for a wide range of threshold-based submodular optimization algorithms where the sampling process is used to determine whether the expectation of the noisy marginal gain is above or below the threshold. Therefore part of our contribution is that we develop and analyze algorithms for different optimization settings of monotone submodular maximization with a matroid constraint (Algorithm 5), and unconstrained non-monotone submodular maximization (Algorithm 4). This former problem setting is especially important for the submodular optimization community because it displays how CS can be used to make the widely used multilinear extension algorithms more efficient.
>
> We now discuss the improvements made by our algorithm CTG over the algorithm EXP-GREEDY of Singla et al. for cardinality constrained monotone submodular maximization.
>
> - Query Complexity. There are two major aspects of the query compleixty of these algorithms: (i) How many times does a marginal gain of $f$ need to be computed over the duration of the algorithm, and (ii) how many samples do we need to take in order to approximate this marginal gain sufficiently well. For (i), our proposed algorithm CTG is proven to be superior to that of Singla et al. by a factor of $O(k)$. In many applications, the input budget $k$ is quite large (even on the order of $n$) and therefore this speedup can make a huge difference. On the other hand, when considering the number of noisy queries for each marginal gain, the proven bounds on the algorithm CTG and EXP-GREEDY are incomparable as they are using completely different noisy sampling strategies, and it would depend on the instance which is superior.
>
> - Improved Robustness. In the EXP-Greedy algorithm, a small $\Delta_{max}$ could directly impact the sample complexity of an entire round of the standard greedy algorithm, where the marginal gains of adding all $n$ elements to the solution set are evaluated and are influenced. In contrast, our algorithm evaluates the marginal gains for different elements independently. As a result, a small value of $\phi$ for one marginal gain does not influence the evaluation of other marginal gains, enhancing robustness.
>
> > Knowing something about the properties of the input instance, how can you estimate the value of $\phi$ without running your algorithm? I am asking this to understand whether your bounds can be used to predict your algorithm's performance. Note that, for example, $\Delta_{max}$ can be estimated in advance based on the properties of the input instance.
>
>  CS effectively lower bounds $\phi$ by $\epsilon$, and therefore this gives us a worst case sample complexity not dependent on $\phi$. In particular, notice in CS we stop sampling once the confidence interval reaches $\epsilon$ (see Figure 1 for illustration), and in Theorems 1 and 2 the left side of the bounds on the number of samples is not dependent on $\phi$.
>
>  On the other hand, notice that $\Delta_{\max}=\min_{i\in[n],i\neq i^*}(\Delta f(S,i^*)-\Delta f(S,i))$ where $\Delta f(S,i^*)=\max_{i\in[n]} \Delta f(S,i)$ is the highest marginal gain. \Delta_{\max} measures the difference of the marginal gain between the element with the highest marginal gain and the element with the second highest marginal gain. The parameter $\phi$ in our algorithm is $\phi=\frac{|\Delta f(S,i)-w|+\epsilon}{2}$. Therefore, in the scenario where we can know something about the properties of the input instance to help estimate $\Delta_{max}$, we should be able to estimate the value of the marginal gain. Notice that the threshold $w$ and parameter $\epsilon$ are known before the algorithm, so we might also be able to estimate $\phi$ in advance.

---

> ### Comment · Reviewer_NxXo · 2024-11-26
>
> I thank you for your explanation. It does address some of my comments. In particular, I did not manage to understand that your algorithm improves the bound on the total number queries by factor of $k$. Your theorems focus on per-iteration bounds which, as you say, are incomparable to Singla et al. I would suggest to point it out more clearly. Even now, I can only see in the appendix B that such an improvement is supposed to happen if $\Delta_{max} \leq O(\epsilon)$, not explaining what will happen if this is not the case. There is at least one other reviewer who had troubles to make sense of your results.
>
> However, looking at the comments of the other reviewers, I have decided to keep my original rating.

---

### Author Response · Authors · 2024-11-23
**Additional response to all reviewers**

Here we address comments made by several reviewers questioning how novel the subroutine Confident Sample (CS) is within submodular optimization (Reviewers jMtB and KUeE). To the best of our knowledge, an adaptive sampling procedure similar to CS has not previously been proposed. What is commonly used in submodular optimization algorithms is what we call a "fixed $\epsilon$-approximation" in the paper (see Section 2). A fixed $\epsilon$-approximation is essentially when one applies a concentration inequality such as Hoeffding's for a fixed number of noisy samples such that the empirical mean of the evaluated random variable $X$ satisfies that $|\hat{X}-\mathbb{E}[X]|\leq\epsilon$.

The fundamental reason this approach is less efficient compared to CS is that we are only interested in determining whether $f(X)$ is approximately above a threshold or not, not in obtaining a precise approximation. In other words, we don't need the guarantee that the $|\hat{X}-\mathbb{E}[X]|\leq\epsilon$ in Hoeffding's inequality; instead, we care about whether $\mathbb{E}[X]\geq w$. Ideally, we would approximate $f(X)$ just finely enough to determine if it's above the threshold. However, this isn't feasible with the fixed $\epsilon$-approximation, because we don't have any prior knowledge of how far $f(X)$ is from the threshold. Thus, we can't determine the required number of samples, and the fixed $\epsilon$-approximation approach requires a single batch of i.i.d. samples.

In contrast, CS uses an adaptive sampling approach where samples are iteratively taken one-by-one until an evolving confidence interval crosses a threshold. The goal is to use fewer samples compared to a fixed $\epsilon$-approximation. While CS might initially seem similar to fixed $\epsilon$-approximation, there are several critical differences that introduce unique technical challenges:

- Fixed $\epsilon$-approximation approach has a batch of samples in which a single application of a concentration inequality is applied in order to approximate $\mathbb{E}[X]$. In contrast, in CS, we apply a concentration inequality after every single sample, and then take a union bound over all the applications. However, this is challenging because we don't know how many samples we will end up taking to approximate the mean value sufficiently well since that depends on the result of the sampling. So we have to carefully design our confidence intervals.
- Fixed $\epsilon$-approximation approach takes a predetermined number of samples, independent of the sampling results. In contrast, the CS algorithm dynamically determines the number of samples based on the outcomes of previous samples. Additionally, CS reuses samples across multiple applications of concentration bounds, enhancing its efficiency.
- In CS, the size of the confidence interval evolves with each additional sample, shrinking as the number of samples increases (see Theorem 1). Additionally, when applying concentration inequalities, the failure probability is adjusted dynamically based on how many samples we've taken so far (Lemma 6). The benefit of the varying failure probability is that the obtained sample complexity $\frac{8R^2}{\phi_X^2}\log\left(\frac{16R^2}{\phi_X^2}\sqrt{\frac{2}{\delta}}\right)$ won't suffer from small values of $\epsilon$.
- In Theorem 2 and 4, we use a combination of Hoeffding and Chernoff that is well-suited to the threshold algorithms, rather than using one or the other. This approach improves the sample complexity from $O(R^2)$ in Theorem 1 to $O(R)$ when $R$ is large.

From a high-level perspective, we want to point out that the sample complexity of submodular optimization problems with stochastic noise usually originated from two reasons: (1). The query complexity of submodular optimization algorithms for the exact oracle value setting (no noise); (2). The sample complexity of estimating the expectation of the submodular objectives. The sample complexity of (2) is usually overlooked and addressed in a simple way. We highlight that the second part, as mentioned by the reviewer JMtB, is usually achieved by only taking a fixed number of samples and is therefore not sample-efficient enough. We minimize the samples required in this process by adaptively deciding when to stop. The only work to the best of knowledge that also uses the adaptive sampling idea is [1] which incorporate the best-arm identification algorithm into the standard greedy approach. However, the algorithm is also not sample-efficient and runtime-efficient. Moreover, our proposed CS algorithms could serve as a general tool for a wide range of threshold-based submodular optimization algorithms where the sampling process is used to determine whether the expectation of the noisy marginal gain is above or below the threshold.

[1] Singla, Adish, Sebastian Tschiatschek, and Andreas Krause. "Noisy submodular maximization via adaptive sampling with applications to crowdsourced image collection summarization."

---

### Author Response · Authors · 2024-11-23
**Response to all reviewers and ACs**

We thank the reviewers for their thoughtful comments. We also appreciate the positive feedback received on the paper, including:
- The problem considered in the paper is interesting and meaningful in the AI and ML community (Reviewers jMtB and KUeE), and the setting is relevant and well-motivated (Reviewers NxXo and KUeE).
- The sample complexity bounds are not trivial (Reviewer KUeE), and the theoretical results of the paper are robust (Reviewer XgDq).
- The theoretical insights are supported by empirical evaluations on real-world datasets (Reviewer XgDq).
- The adaptive sampling in the paper is different from existing fixed-precision methods (see Section 2 for a definition of fixed $\epsilon$-approximation) (Reviewer XgDq).
- The algorithms are a competitive alternative to existing ones (Reviewer XgDq).
- The paper is well-written and structured (Reviewer jMtB).

Several reviewers made comments questioning how our sampling subroutine Confident Sample (CS) is different compared to sampling approaches existing in the submodular optimization literature (Reviewers jMtB and KUeE). We address these remarks in a separate comment at https://openreview.net/forum?id=vtCkb4KJxr&noteId=7unTbMJ62F.

We have uploaded a revised version of the manuscript, incorporating reviewer feedback with changes highlighted in blue. The updates include:
- A discussion of the technical challenges and a comparison with the fixed $\epsilon$-approximation approach (Appendix C.1).
- Detailed description and explanation of the results of sample complexity. (Sections 3 and 4).

We believe these revisions have improved the paper and welcome further feedback. Below are our detailed responses to individual reviewer comments.

---

### Meta-Review · Area_Chair_N6eK · 2024-12-22

**Metareview:**

The paper studies the problem of maximizing a submodular function in the setting where we only have access to noisy evaluations of the marginal gains. Specifically, the noisy evaluations are assumed to be iid samples from a sub-Gaussian distribution. This setting mirrors exactly that of a prior work (Singla et al., 2016), who studied the cardinality constrained problem and proposed an algorithm based on the standard Greedy algorithm. The current paper considers the cardinality constrained problem as well as a more general matroid constraint and unconstrained non-monotone maximization. The current work shows how to implement the well-known descending thresholds Greedy algorithm for submodular maximization using noisy evaluations, and achieve a smaller sample complexity, particularly in practice. The main contribution is a procedure called Confident Sample (CS) that implements the main step of the algorithm using noisy samples: given an element and a threshold, determine whether the marginal gain of the element on top of the current solution exceeds the threshold. Instead of taking a fixed number of samples in order to estimate the marginal gain sufficiently accurately, the CS algorithm only takes enough samples to determine whether the gain is above the threshold. This is beneficial in practice, since we only need to take the worst-case number of samples only when the marginal gain is very close to the threshold. The paper gives theoretical guarantees that quantify the improvement in the sample complexity parameterized by how far away the marginal gain is from the threshold, and it empirically evaluates the improvement on data summarization instances.

The problem studied is well motivated and relevant for machine learning applications. The proposed algorithm has the potential to reduce the number of samples taken in practice compared to other approaches based on worst-case concentration results, which give pessimistic bounds when the marginal gain is far from the threshold. A major weakness noted by the reviewers is that the overall contribution is modest. The main contribution is the application of the CS algorithm to the well-known descending thresholds algorithm. The CS algorithm itself is novel and the analysis poses some challenges but, as Reviewers JMtB and KUeE noted, the novelty in the algorithm design and the analysis is limited.

**Additional Comments On Reviewer Discussion:**

The author response clarified several of the reviewers' questions regarding the improvements over the prior work as well as the fit of this work to ICLR. The authors revised the paper to add additional discussion and comparison to prior work. Following the discussion, the main concern that the contribution is modest still remained.

---

### Decision · Program_Chairs · 2025-01-22

Reject